# Empowering Small VLMs to Think with Dynamic Memorization and Exploration

**Jiazhen Liu**[iD]**, Yuchuan Deng**[iD]**, and Long Chen**[*]
The Hong Kong University of Science and Technology
https://github.com/HKUST-LongGroup/DyME

## Abstract

Small-scale Vision–Language Models (SVLMs) are exceptionally well-suited for proprietary tasks. Equipping them with thinking capabilities is a critical step to enhance their performance and reliability in these specific domains. However, existing training paradigms, including Supervised Fine-Tuning (SFT) and Reinforcement Learning with Verifiable Reward (RLVR), impose substantial demands on the base VLM, exceeding the capacity of SVLMs. Consequently, directly applying these paradigms to SVLMs fails to instill the desired thinking abilities. A natural solution is to combine SFT and RLVR, leveraging their complementarity to reduce the dependence on model capacity. Yet the core challenge lies in managing the inherent trade-off: excessive reliance on SFT can force the model to memorize pseudo thinking traces, while over-emphasizing RLVR can lead to unstable exploration (*i.e.*, advantage collapse). To address this, we propose DyME, a novel training paradigm that **Dy**namically selects between **M**emorization (via SFT) and **E**xploration (via RLVR) at each optimization step. By ensuring that every update contributes to the trade-off, DyME serves as a robust, standalone strategy that stabilizes SVLM learning. Complementing this paradigm, we further introduce a synergistic *Visual Supervision* mechanism (comprising a visual checker and refiner) designed to inject dynamically enhanced, image-grounded guidance during optimization. Extensive experiments across diverse domains demonstrate that DyME consistently achieves this balance, and thus delivers substantial performance improvements on specialized tasks. These results establish DyME as a practical and effective solution for empowering SVLMs with reliable thinking capabilities.

## 1 Introduction

Equipping Vision–Language Models (VLMs) with thinking capabilities is a pivotal step that moves them beyond recognition toward reasoning. Recent studies have advanced this goal through specialized training, achieving strong results on a spectrum of visual tasks, from recognition-intensive applications like grounding (Lai et al., 2025; Liu & Chen, 2025; Peng et al., 2025; Liu et al., 2025c;a) to reasoning-intensive challenges such as chart understanding (Zhang et al., 2025a; Xia et al., 2024) and geometric problem solving (Shen et al., 2025; Chen et al., 2025b; Xia et al., 2025). While this progress is significant, the success of these approaches is contingent upon the base VLM possessing strong foundational capabilities, namely, sufficient capacity and robust instruction adherence (Yang et al., 2025a). In practice, only a handful of VLMs meet these prerequisites, presenting a significant challenge for Small-scale VLMs (SVLMs) which struggle to develop thinking capabilities under existing training paradigms.

To contextualize this limitation, we briefly review the two dominant paradigms, both of which are primarily tailored for Large-scale VLMs (LVLMs). **1) Supervised Fine-Tuning (SFT) on Chain-of-Thought (CoT) data** (Xu et al., 2024; Li et al., 2024b; Xia et al., 2025; Gao et al., 2025): VLMs are supervised to memorize predefined thinking patterns from large-scale CoT annotations. Since CoT data are often verbose and contain much vision-irrelevant content, models must possess sufficient capacity to absorb long textual content without compromising visual grounding (Marafioti et al.,

---

[*]Corresponding author (longchen@ust.hk)

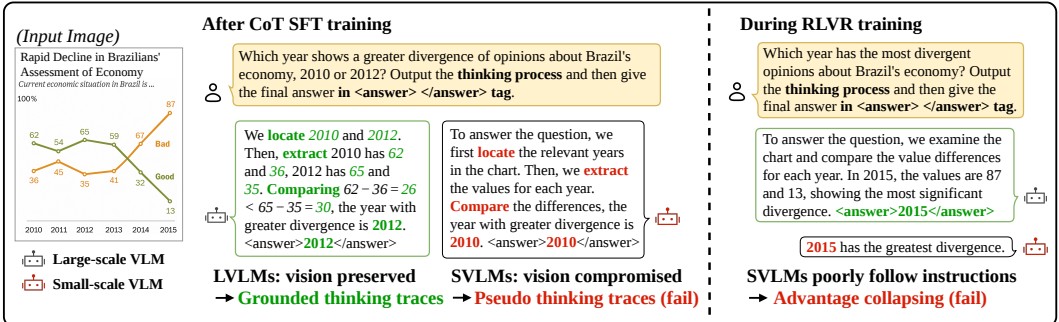

a) SFT and RL paradigms fail to enable SVLMs to think.

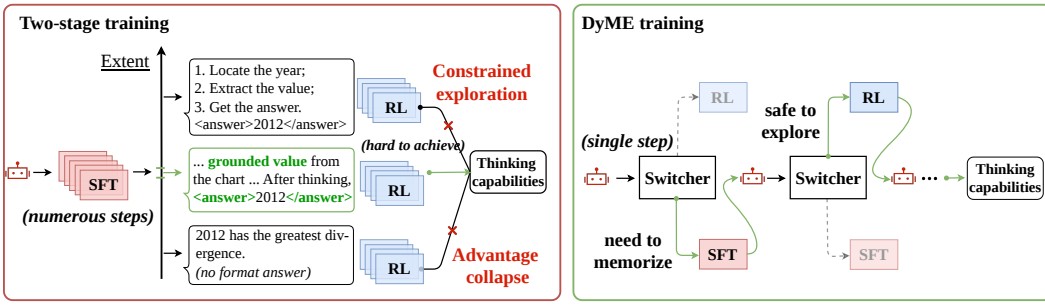

b) Two-stage training *vs.* DyME

Figure 1: **Training paradigms for enabling VLM thinking**. The LVLM is Qwen2.5-VL-32B (Bai et al., 2025) and the SVLM is SmolVLM-500M (Marafioti et al., 2025). (a) Existing paradigms are effective for LVLMs but unsuitable for SVLMs. (b) The two-stage training paradigm (SFT → RL) faces a challenging trade-off. Our proposed DyME dynamically balances this trade-off.

2025). This capability gap is illustrated in Fig. 1a: After SFT, LVLMs can generate grounded thinking traces with accurate intermediate values (in green), while SVLMs cannot. **2) Reinforcement Learning with Verifiable Reward (RLVR)** (Zhang et al., 2025a; Chen et al., 2025b; Peng et al., 2025; Shen et al., 2025): on the other hand, promotes exploration of thinking patterns rather than imitations. In this paradigm, VLMs are instructed to generate a thought process followed by a strictly formatted answer (*e.g.*, enclosed in tags). This format enables verifiable rewards to reinforce correct generations and penalize incorrect ones. Owing to its reliance on instruction adherence, this approach is practical primarily for strong VLMs that can reliably generate structured outputs.

Consequently, both established paradigms are inadequate for instilling thinking in SVLMs. The extremely limited capacity (*e.g.*, under 1B parameters) of SVLMs renders the SFT paradigm ineffective, as a high volume of textual information in CoT data can overwhelm the capacity (Marafioti et al., 2025; Chen et al., 2025a). Moreover, the limited instruction adherence of SVLMs frequently results in unverifiable outputs (Chu et al., 2025; Guo et al., 2025), precipitating advantage collapse during RLVR. We quantitatively verify these limitations (*cf.*, Fig. 2): both SFT and RLVR paradigms indeed impair the performance.

Considering that SVLMs offer high efficiency and are crucial for deployment on edge devices (Marafioti et al., 2025), enabling them to think addresses a strong practical demand. Thinking enhances the reliability and performance of vision tasks (Zhang et al., 2025a), and task-specific SVLMs provide a compelling alternative to LVLMs in resource-constrained settings. This motivates the development of a new training paradigm that empowers SVLMs with thinking capabilities, at least for specialized tasks.

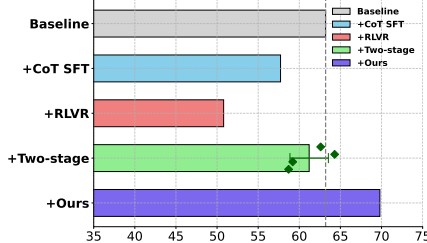

Figure 2: **Performance of SmolVLM-500M (Marafioti et al., 2025) on ChartQA (Masry et al., 2022)**. Existing paradigms degrade performance, whereas DyME yields improvements.

A promising solution is to fuse SFT and RLVR, as a well-calibrated **trade-off** can lower the high demands on the base model (DeepSeek, Inc., 2025; Yan et al., 2025): SFT encourages the model to memorize verifiable thinking patterns to prevent advantage collapse, while RL forces exploration to prevent rigid templates from overwhelming the model's capacity. The central challenge, however, is that SVLMs struggle to achieve this balance. Existing hybrid methods, like two-stage training (Chen et al., 2025a; Chu et al., 2025) or annealed SFT losses[1] (Zhang et al., 2025b), rely on a static trade-off governed by hyperparameters set empirically. This rigidity is the critical flaw because the minimal capacity of SVLMs means the window for a successful static balance is incredibly narrow, making failure almost inevitable (*cf.* Fig. 1b). Our repeated trials with two-stage training confirmed this issue, with performance often falling below the baseline (*cf.* Fig. 2).

SVLMs therefore require a more intelligent paradigm to navigate this trade-off. To this end, we propose DyME (**Dy**namic **M**emorize–**E**xplore), which integrates SFT and RLVR through a **dynamic** switching mechanism. As illustrated in Fig. 1b, DyME assesses the model's generation at each step and adapts its training mode accordingly. When the model fails to follow instructions, it switches to a memorization mode (SFT) to guarantee stable optimization signals. Conversely, for valid generations, it engages an exploration mode (RLVR) to encourage diverse and grounded thinking. This state-driven approach ensures memorization and exploration are always complementary, dynamically maintaining the delicate trade-off. While this dynamic switching alone guarantees training stability, we further maximize the model's potential by incorporating a synergistic *Visual Supervision* mechanism. This module facilitates an adaptive interaction: the CoT ground-truth guides the scoring of exploration (via a visual checker), while successful exploration traces dynamically refine the CoT ground-truth (via a visual refiner).

The aforementioned design makes DyME a highly effective paradigm for empowering thinking in SVLMs for specific tasks. We validate this across three diverse domains, ranging from recognition-intensive tasks (medical VQA) to reasoning-intensive challenges (chart understanding and geometric problem solving). Remarkably, using only a few thousand training samples, DyME achieves substantial performance gains, enabling it to match or even surpass several LVLMs. Our primary contributions are as follows:

1. We propose DyME, the first training paradigm that equips SVLMs with thinking capabilities, substantially reducing reliance on the base VLM's initial capacity.
2. Through dynamic switching and synergistic supervision, DyME alleviates pseudo thinking traces and advantage collapse in SVLMs, yielding image-grounded thinking and consistent performance improvements.
3. We demonstrate the effectiveness and practicality of DyME across three diverse domains, each consistently showing substantial performance gains with only a few thousand training samples.

## 2 RELATED WORK

**Vision-Language Models.** Modern VLMs, such as LLaVA (Liu et al., 2024a) and Qwen-VL (Bai et al., 2023), have demonstrated remarkable capabilities across a wide array of vision tasks. However, their substantial parameter counts and computational demands restrict their use in resource-constrained environments like edge devices. This has motivated a growing interest in SVLMs designed for efficiency (Zhou et al., 2024; Marafioti et al., 2025; Korrapati, 2024). Although works like TinyLLaVA (Zhou et al., 2024) and SmolVLM (Marafioti et al., 2025) have shown that carefully designed SVLMs can achieve competitive performance, they exhibit a critical weakness. Recent studies highlight that their performance degrades significantly on tasks requiring complex, multi-step instruction following, indicating a gap in their compositional understanding and general reasoning abilities (Albalak et al., 2022; Ghosh et al., 2024; Liu et al., 2025b).

**Empowering Thinking Capabilities in VLMs.** Recent advances in LLM thinking (*e.g.*, GPT-o1 (OpenAI, 2024), DeepSeek-R1 (Guo et al., 2025)) have motivated efforts to equip VLMs with similar capabilities via dedicated training paradigms.

*SFT on CoT data* (Xu et al., 2024; Xia et al., 2024; 2025; Gao et al., 2025; Yang et al., 2025b). This paradigm leverages large-scale CoT supervision to teach models to memorize and generalize thinking patterns. Multimodal-CoT (Zhang et al., 2023) was an early attempt using fused visual–text

---

[1]See the supplementary material for further comparison.

inputs, but its small scale data limited genuine thinking. Subsequent works highlight the role of scale: G-LLaVA (Gao et al., 2025) constructs 170K geometry-specific CoT samples; ChartVLM (Xia et al., 2024) compiles a large chart corpus; and LLaVA-CoT (Xu et al., 2024) as well as R1-OneVision (Yang et al., 2025b) curate diverse, structured CoT data through large-scale prompt engineering. These approaches face long inputs, requiring large VLMs that can process rich textual information while preserving visual grounding (Marafioti et al., 2025; Zhai et al., 2023).

*RL with Verifiable Reward (RLVR)* (Zhang et al., 2025a; Chen et al., 2025b; Peng et al., 2025; Shen et al., 2025; Liu et al., 2025c). RLVR adopts a distinct paradigm that elicits thinking through autonomous exploration with minimal external supervision. The popularly used algorithm is Group Relative Policy Optimization (GRPO), introduced by DeepSeek-Math (Shao et al., 2024), which exploits models' ability to produce structured outputs that separate thinking from final answers. It leverages rule-verifiable data to optimize high-scoring generations, while light SFT is employed for cold-start when the output structure is unclear. This paradigm has been extended to VLMs in several works. R1-V (Chen et al., 2025b) applies GRPO to VLMs, enabling thinking in tasks such as counting and geometry. LMM-R1 (Peng et al., 2025) introduces a two-stage pipeline that transfers textual thinking into multimodal learning. VisualRFT (Liu et al., 2025c) and R1-VL (Zhang et al., 2025a) incorporate vision-specific rewards to guide fine-grained, visually grounded optimization. Since GRPO depends on models' initial structured thinking ability, these methods typically build on strong VLMs, such as the Qwen-VL series (Bai et al., 2025).

*Hybrid Training Paradigms* (Chu et al., 2025; Yan et al., 2025; Zhang et al., 2025b). To harness the complementary strengths of SFT and RL, researchers have also investigated hybrid paradigms. A common approach is a two-stage training process (Chu et al., 2025) that first uses SFT to teach the model the desired output format, followed by RL for exploration. Although intuitive, this method is highly sensitive to the amount of SFT, a parameter that is particularly challenging to tune for SVLMs, as these smaller models can easily become trapped in suboptimal states. Alternative strategies attempt to continuously blend SFT with RL, for instance, by incorporating SFT as an annealed auxiliary loss (Zhang et al., 2025b) or by managing its influence with an empirical shaping function (Yan et al., 2025). However, all these strategies ultimately rely on an empirically determined balance between the two paradigms. This rigidity represents a critical flaw when applied to SVLMs. The absence of adaptive control over the SFT weight renders these methods brittle and unreliable.

Thus, existing paradigms are not directly transferable to SVLMs due to their inherent limitations in model capacity and instruction-following ability. This highlights the need for a novel training paradigm that imposes minimal requirements on the base VLM.

## 3 APPROACH

### 3.1 PRELIMINARIES

We first briefly recap the two training paradigms (SFT and RLVR) that underlie our method. Let $\mathcal{D} = \{(x_i, y_i)\}_{i=1}^N$ be the training set, where $x$ denotes the input (*e.g.* an image-instruction pair) and $y$ the desired output. The model defines a conditional distribution $p_\theta(y \mid x)$ with parameters $\theta$.

**Supervised Fine-Tuning (SFT).** For each training pair $(x, y)$ in $\mathcal{D}$, SFT updates the model by minimizing the negative log-likelihood (cross-entropy) of the desired output $y$ under the conditional distribution $p_\theta(y \mid x)$:

$$\mathcal{L}_{\text{SFT}}(\theta) = -\mathbb{E}_{(x,y)\sim\mathcal{D}}\big[\log p_\theta(y \mid x)\big]. \tag{1}$$

This teacher-forcing loss allows models to *memorize* extensive training examples, compelling the model to absorb this knowledge.

**Group Relative Policy Optimization (GRPO).** GRPO is an RL algorithm that *explores* open-ended generation by comparing candidate outputs within a group. For each input $x$, the policy $p_\theta$ samples a set $\{\tilde{y}^k\}_{k=1}^K$; a reward function $r_a(\tilde{y}^k)$ is computed based on the correctness of the output answer, and each sample's advantage $A$ is measured relative to the other group members:

$$A(\tilde{y}^k) = \frac{r_a(\tilde{y}^k) - \bar{r}_a}{\sigma + \varepsilon}, \quad \bar{r}_a = \frac{1}{K}\sum_{j=1}^K r_a(\tilde{y}^j), \quad \sigma = \sqrt{\frac{1}{K}\sum_{j=1}^K (r_a(\tilde{y}^j) - \bar{r})^2}, \tag{2}$$

where $\varepsilon$ is a small constant for numerical stability. The policy then updates its parameters by minimizing the following loss, regularised by a KL constraint:

$$\mathcal{L}_{\text{GRPO}}(\theta) = -\mathbb{E}_{x \sim \mathcal{D}} \, \mathbb{E}_{\tilde{y} \sim p_\theta} \Big[ \min\big(r_\theta(x, \tilde{y}) \, A(\tilde{y}), \text{clip}\big(r_\theta(x, \tilde{y}); 1 - \epsilon, 1 + \epsilon\big) A(\tilde{y})\big) \Big]$$

$$+ \beta \, D_{\text{KL}}\big[ p_\theta(\cdot \mid x) \, \| \, p_{\text{ref}}(\cdot \mid x) \big], \quad \text{where} \quad r_\theta(x, \tilde{y}) \; = \; \frac{p_\theta(\tilde{y} \mid x)}{p_{\text{old}}(\tilde{y} \mid x)}. \tag{3}$$

The clip and KL terms work together to keep each update close to safe regions of the parameter space: the clip gate limits step size around the rollout policy $p_{\text{old}}$, while the KL term ($\beta D_{\text{KL}}$) tethers the policy to the reference $p_{\text{ref}}$ (typically the initial model).

**Gradient Compatibility of SFT and GRPO.** Below, we reveal that the optimization objectives of SFT and GRPO are formally equivalent, with the former targeting the ground-truth data distribution and the latter an internal one.

The gradient of the SFT loss is straightforward:

$$\nabla_\theta \mathcal{L}_{\text{SFT}}(\theta) = -\mathbb{E}_{(x,y) \sim \mathcal{D}} \left[ \nabla_\theta \log p_\theta(y \mid x) \right]. \tag{4}$$

Similarly, the GRPO gradient (ignoring clipping and any KL-penalty) can be written as

$$\nabla_\theta \mathcal{L}_{\text{GRPO}}(\theta) = -\mathbb{E}_{\substack{x \sim \mathcal{D}, \\ \tilde{y} \sim p_{\text{old}}(\cdot | x)}} \left[ r_\theta(x, \tilde{y}) \, A(\tilde{y}) \, \nabla_\theta \log p_\theta(\tilde{y} \mid x) \right]. \tag{5}$$

This comparison shows that the SFT gradient is a special case of the GRPO gradient, obtained when the ground-truth sample is used with unit advantage. This equivalence enables a unified loss that balances external imitation (SFT) with internal refinement (GRPO). Achieving this fusion requires dynamically weighting the two signals (§3.2) and ensuring stylistic consistency between external ground-truth and self-generated outputs (§3.3).

## 3.2 DYNAMIC MEMORIZE–EXPLORE (DYME)

To realize this complementarity, we propose the **Dy**namic **M**emorize–**E**xplore (DYME) paradigm, which adaptively switches between SFT and GRPO at each training step. In the following, we first outline the overall pipeline and then elaborate on the optimization procedures for each mode.

**Overall.** As shown in Fig. 3a, each training step begins with an input $x = (I, q)$, where $I$ is the image and $q$ is an instruction. The policy SVLM $p_\theta$ generates $K$ responses $\{\tilde{y}^k\}_{k=1}^K$. Each response is parsed into a thinking trace and a final answer, which is then verified for correctness using predefined rules. The verification results fall into two categories: either all responses are incorrect (including those that fail to parse), or at least one is correct. **The decision rule:** if at least one response is correct, the model proceeds with GRPO-based exploration; otherwise, it falls back to SFT-based memorization. Formally, the training mode is switched as:

$$\text{mode}(x) = \begin{cases} \text{GRPO}, & \text{if } \max_k r_a(\tilde{y}^k) = 1, \\ \text{SFT}, & \text{otherwise}, \end{cases} \tag{6}$$

where $r_a(\tilde{y}^k) \in \{0, 1\}$ indicates whether $\tilde{y}^k$ passes rule-based verification. Though simple, this decision rule is highly effective. When all responses are incorrect, the answer rewards are essentially all zero and the normalized advantages become dominated by noise, making GRPO updates for a small SVLM unstable. In this regime, falling back to SFT provides a low-variance, ground-truth guided gradient. Conversely, the appearance of at least one correct response indicates that the current policy has already discovered a feasible solution for this input, so GRPO can safely exploit the relative advantages to drive exploration.

**GRPO Mode.** DYME introduces a key refinement to the original GRPO: beyond the answer reward $r_a$, it incorporates an auxiliary reward $r_t$ for thinking traces. This reward is computed by evaluating the generated traces against expected thinking patterns (*e.g.*, via token-level F1 score ground-truth comparison), promoting structured thinking.

Given these rewards, we update the policy using a modified GRPO objective. Unlike the standard formulation (Eqs. 2 & 3), we omit the KL penalty and clipping terms, as the dynamic integration of

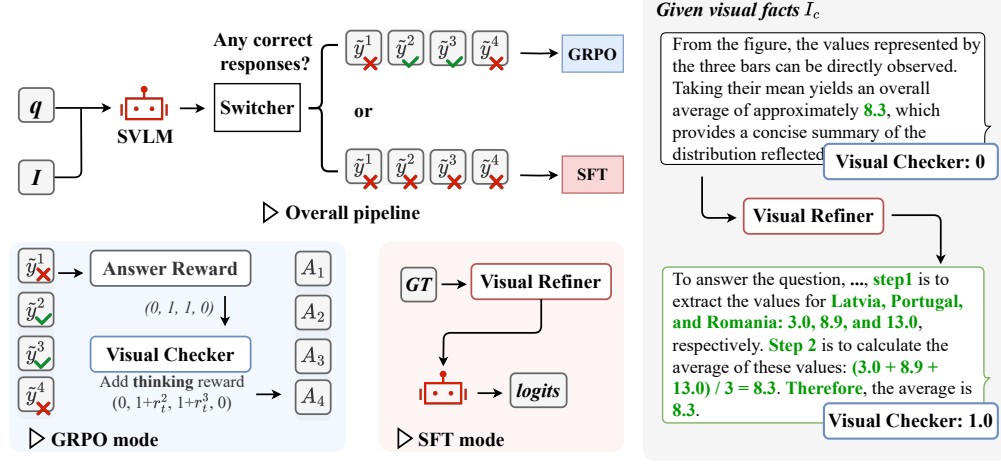

(a) The pipeline for DyME.

(b) Visual refiner and checker.

Figure 3: **Workflow and module components of DyME.** At each training step, DyME dynamically switches between memorization (via SFT) and exploration (via GRPO) modes based on its generations. Visual supervision is introduced through the visual refiner and visual checker. The refiner enhances the targets for memorization by incorporating richer visual elements (green), while the checker rewards the thinking context generated based on their visual relevance.

SFT already stabilizes training. This avoids overly conservative updates and yields a cleaner gradient form, enabling smoother alignment between SFT and GRPO:

$$\tilde{\mathcal{L}}_{\text{GRPO}}(\theta) = -\mathbb{E}_{x\sim\mathcal{D}}\,\mathbb{E}_{\tilde{y}\sim p_\theta(\cdot|x)}\left[r_\theta(x,\tilde{y})\,A(\tilde{y})\right], \tag{7}$$

where $A(\tilde{y}^k)$ is the group-normalized advantage calculated from the combined answer ($r_a$) and thinking ($r_t$) rewards, and $r_\theta(x,\tilde{y}^k) = \frac{p_\theta(\tilde{y}|x)}{p_{\text{old}}(\tilde{y}|x)}$ is the importance sampling ratio.

**SFT Mode.** When training falls back to SFT, the model is optimized toward the ground-truth response $y$ using the standard supervised loss (Eq. 1). This ensures that whenever the model fails to explore effectively, it receives a stable, ground-truth-guided gradient update to correct its behavior.

**DyME Objective.** The final loss dynamically combines the two objectives based on response correctness:

$$\mathcal{L}_{\text{DyME}}(\theta) = \mathbb{1}\left[\max_k r_a(\tilde{y}^k) = 1\right]\cdot\tilde{\mathcal{L}}_{\text{GRPO}}(\theta) + \left(1 - \mathbb{1}\left[\max_k r_a(\tilde{y}^k) = 1\right]\right)\cdot\mathcal{L}_{\text{SFT}}(\theta), \tag{8}$$

where $\mathbb{1}[\cdot]$ is the indicator function, returning 1 if the condition holds, 0 otherwise.

### 3.3 VISION SUPERVISION

**DyME with Visual Supervision.** While the aforementioned **Pure DyME** (using standard $r_t$ and static ground-truth) already guarantees training stability through its dynamic switching mechanism, we can further exploit this dynamic nature to maximize performance. Specifically, the switching mechanism allows us to tailor the supervision signals at each optimization step: refining the reward during exploration and enhancing the ground-truth during memorization. To this end, we introduce a *checker–refiner* framework (*cf.* Fig. 3b), which constitutes the **Full DyME**.

This framework reorganizes the ground-truth to adhere to a predefined structure, crucially transforming it into a grounded thinking trace. The refiner restructures the external ground-truth into structured, visually grounded responses, while the checker evaluates self-generated outputs for their structural organization and coverage of visual content. We refer to the resulting supervision signals collectively as *vision supervision*. The implementation is carried out via LLM-based prompt engineering.

**Visual Facts $I_c$** are central to realizing vision supervision. They are defined as fine-grained visual components extracted from each image, including objects, attributes, and states. These elements play

a dual role: they provide evidence for evaluating generations against the image and serve as building blocks for constructing complete ground-truth responses.

**Visual Checker.** The visual checker evaluates responses along two dimensions: (i) whether the output contains sufficient correct visual elements compared to $I_c$, and (ii) whether it aligns stylistically with provided examples. These examples may be manually defined or extracted from the SFT ground-truth.

**Visual Refiner.** The refiner produces visually grounded responses for SFT by leveraging the model's validated explorations. High-scoring traces identified by the visual checker are stored in a dynamic example pool. An LLM then draws from this pool to generate ground-truth responses, integrating structural templates with visual facts from $I_c$ and referencing the collected examples.

In essence, the acquisition of Visual Facts, the evaluation by the Visual Checker, and the synthesis by the Visual Refiner are all implemented via structured prompt engineering using Qwen2.5-14B. Please refer to the Supplementary Materials for the full prompts used in our pipeline.

## 4 EXPERIMENTS

To rigorously evaluate DyME, we structure our experiments into two parts: (1) **Algorithmic Validation**, where we evaluate "Pure DyME" in a controlled setting using offline data to isolate the contribution of our dynamic switching mechanism; and (2) **System Effectiveness**, where we evaluate the full DyME pipeline (with Visual Supervision) across diverse domains to demonstrate its practical capability in empowering SVLMs.

### 4.1 PART I: ALGORITHMIC VALIDATION (PURE DyME)

**Setup.** Since SVLMs lack intrinsic reasoning capabilities and cannot autonomously discover complex reasoning paths, pre-constructed CoT data is a mandatory prerequisite for all training paradigms. We therefore evaluated all methods on ChartQA (Masry et al., 2022) using LLaVA-OV-S (Li et al., 2024a), the 0.5B variant, with three pre-constructed CoT datasets of varying qualities: **Low (Undesigned)** containing unstructured traces ($\sim$80 words); **Medium (Standard)** consisting of semi-structured traces ($\sim$89 words) from Qwen2.5-14B; and **High (Premium)** comprising highly structured traces ($\sim$142 words) from GPT-4o. Following established protocols (Liu et al., 2023; Masry et al., 2022), we report *relaxed correctness*, which allows a 5% tolerance for numerical answers.

We present a threefold evaluation to validate data robustness, design optimality, and generalization:

**(1) Robustness to Data Quality.** Table 1 (a) demonstrated DyME's superiority. On *Low* quality data, Pure DyME (61.9%) significantly outperforms the unstable Two-stage baseline (57.6%). Remarkably, using only *Medium* data, it surpasses the SFT baseline trained on premium *High* (GPT-4o) data (61.6%). This confirms that DyME acts as a robust student, effectively maximizing data efficiency.

**(2) Optimality of Binary Switching.** To validate our binary design, we compared it against three alternative switching heuristics in Table 1 (b): (i) *Reward Thresholding*, which switches to RL only if the batch average reward exceeds a threshold $t$; (ii) *SFT Annealing*, which applies a weighted SFT loss alongside RL at every step; and (iii) *SFT Budget*, which performs focused SFT updates on accumulated failure cases (hard mining).

**Results:** *Reward Thresholding* proves brittle, collapsing at suboptimal thresholds ($t = 0.5, 52.4\%$). *SFT Annealing* incurs a heavy computational tax ($+25\%$) due to the auxiliary SFT gradient calculation. *SFT Budget* yields inferior results (59.6%) as overwhelming the model with concentrated failures destabilizes learning. In contrast, DyME's binary switch is parameter-free, efficient, and empirically optimal (64.9%).

**(3) Mechanism Generality.** Going beyond the primary setup, while DyME is primarily tailored for SVLMs, we verify the universality of its core switching mechanism (see Supplementary). In the text-only domain, it boosts the small-scale Qwen2.5-0.5B on GSM8K (Cobbe et al., 2021) to 55.3% (+5.8% over GRPO), confirming DyME is an effective paradigm for empowering thinking in small-parameter models regardless of modality. Moreover, the paradigm scales effectively: on the stronger Qwen2.5-VL-7B, it further improves ChartQA performance to 89.6% (+2.3%).

Table 1: **Algorithmic Validation of Pure `DyME`.** (a) `DyME` outperforms SFT and Two-stage variants (w/ and w/o KL penalty) across all data qualities. (b) The binary switch is more robust and efficient than soft or hard-mining alternatives (evaluated on Medium data).

| (a) Robustness across Data Quality | | | | (b) Switching Strategy Ablation | | | |
|---|---|---|---|---|---|---|---|
| **Method** | **Low** | **Medium** | **High** | **Strategy** | **Hyperparam.** | **Acc.** | **Cost** |
| SFT | 50.5 | 57.8 | 61.6 | Reward Threshold | $t = 0.5/0.8/0.9$ | 52.4/64.1/63.4 | None |
| Two-stage | 57.6 | 59.9 | 54.5 | SFT Annealing | Cosine | 64.0 | +25% |
| Two-stage (w/ KL) | 55.4 | 60.8 | 62.7 | SFT Budget | Hard Mining | 59.6 | Budget-dep. |
| **Pure `DyME`** | **61.9** | **64.9** | **68.5** | **Binary Switch (Ours)** | – | **64.9** | **Baseline** |

## 4.2 PART II: SYSTEM EFFECTIVENESS (FULL `DyME`)

Having validated the algorithmic core, we now evaluate the Full `DyME` pipeline, augmented with Visual Supervision, across three diverse domains: Medical VQA, Chart Understanding, and Geometry. Each followed the evaluation protocols of prior work (Zong et al., 2024).

**Setup & Source of $I_c$.** Unlike Part I, here we activate the Visual Supervision module to enable the full online loop. Crucially, to demonstrate `DyME`'s capability to bootstrap from raw signals, we utilize the "Undesigned" CoT data (defined in §4.1) derived from SLAKE (Liu et al., 2021), ChartQA (Masry et al., 2022), and Geo170K (Gao et al., 2025) as the common training source for all methods. Acquiring the necessary visual facts ($I_c$) is a fully automated process: we leverage standard domain tools (*e.g.*, BiomedGPT (Zhang et al., 2024a) for medical, DePlot (Liu et al., 2023) for charts) or prompt generalist LLMs (*e.g.*, Qwen2.5 (Team, 2024)) to parse images into structured textual descriptions. The automated pipeline and prompts are included in the supplementary.

**Evaluation Protocol.** We used official train-test splits for SLAKE (Accuracy/Recall) and ChartQA (Relaxed correctness). For Geometry, since Geo170K (Gao et al., 2025) provides no test set, we evaluated Accuracy on MathVerse (Zhang et al., 2024b), consistent with Zong et al. (2024).

### 4.2.1 MAIN RESULTS

**`DyME` *vs.* Existing Training Paradigms**. The comprehensive results in Table 2 show that `DyME` consistently delivers substantial gains. Notably, after training with `DyME`, SmolVLM improves from 49.9 to 55.6 (+5.7), LLaVA-OV-S from 50.7 to 55.4 (+4.7), and InternVL2-S from 56.3 to 58.1 (+1.8). In contrast, existing paradigms tend to degrade performance (*e.g.*, SFT lowers SmolVLM to 44.1), validating our analysis that SFT yields pseudo thinking traces and GRPO faces advantage collapse (*cf.* Fig. 4).

`DyME` effectively mitigates these issues. It promotes grounded traces that are concise yet informative (*cf.* Fig. 5), aligning well with the limited capacity of SVLMs. Importantly, `DyME` places minimal demands on the base model: even SmolVLM (0.5B) achieves substantial gains, and it still delivers improvements (+2.6%) on extensively pretrained models like InternVL2-S. We further corroborated these findings through manual inspection, as detailed in the Supplementary Material.

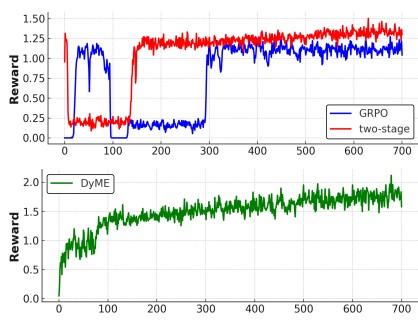

Figure 4: **Training rewards.** GRPO and two-stage training suffer from severe advantage collapse.

**Matching the Efficacy of GPT-4o Supervision with Open-Source Models.** Comparing results between Part I and Part II reveals a crucial finding: LLaVA-OV-S trained with the full `DyME` pipeline (using the accessible Qwen2.5-14B) achieves 67.5% (Table 2). This effectively matches the performance of Pure `DyME` trained on expensive GPT-4o data (68.5%, *cf.* Table 1). This proves that full `DyME` allows open-source supervision to achieve training outcomes comparable to those derived from top-tier proprietary models, eliminating the need for expensive data annotation.

**`DyME`-trained SVLMs Can Be Competitive with LVLMs.** We ensured fairness by exposing all baselines to our training data. As shown in Table 2, SVLMs trained with `DyME` can surpass stronger

Table 2: **Comparisons across three domains: medical VQA, chart understanding, and geometry solving.** The evaluation follows the VLMEvalKit framework (Duan et al., 2024). For SVLMs, existing training paradigms degrade their performance, whereas `DyME` consistently brings improvements. The best performance achieved by each SVLM is highlighted in bold, with the relative improvement also indicated. Notably, after being trained with `DyME`, SVLMs achieve performance comparable to that of MoVA (underlined).

| Model | ViT | LLM | Medical | Chart | Geometry | Avg. |
|---|---|---|---|---|---|---|
| **LVLMs** | | | | | | |
| LLaVA-Med (Li et al., 2023) | CLIP-ViT-L/14 | Vicuna-7B | 64.3 | – | – | – |
| Cambrian-1 (Tong et al., 2024) | Hybrid-3B | Llama3-8B | – | 72.6 | 22.0 | – |
| LLaVA-1.5 (Liu et al., 2024a) | CLIP-ViT-L/14 | Vicuna-7B | 69.4 | 17.8 | – | – |
| LLaVA-1.6 (Liu et al., 2024b) | CLIP-ViT-L/14 | Vicuna-7B | 78.2 | 49.2 | 13.4 | 47.0 |
| MoVA (Zong et al., 2024) | Hybrid-3B | Vicuna-7B | 74.5 | 68.3 | 19.7 | 54.2 |
| LLaVA-OV-L (Li et al., 2024a) | SigLIP-SO400M | Qwen2-7B | 75.7 | 80.9 | 24.5 | 60.4 |
| InternVL2-L (Chen et al., 2024) | InternViT-300M | InternLM2.5-7B | 80.2 | 82.1 | 37.3 | 66.5 |
| **SVLMs** | | | | | | |
| SmolVLM (Marafioti et al., 2025) | SigLIP-93M | SmolLM2-360M | 72.1 | 63.2 | 14.6 | 49.9 |
| + CoT SFT | SigLIP-93M | SmolLM2-360M | 60.1 | 57.7 | 14.5 | 44.1 |
| + GRPO | SigLIP-93M | SmolLM2-360M | 61.1 | 53.8 | 17.1 | 44.0 |
| + Two-stage | SigLIP-93M | SmolLM2-360M | 59.4 | 60.1 | 16.7 | 45.4 |
| **+ DyME** | SigLIP-93M | SmolLM2-360M | **78.1** (+6.0%) | **69.7** (+6.5%) | **18.9** (+4.3%) | **55.6** (+5.7%) |
| LLaVA-OV-S (Li et al., 2024a) | SigLIP-400M | Qwen2-0.5B | 74.9 | 61.4 | 15.9 | 50.7 |
| + Two-stage | SigLIP-400M | Qwen2-0.5B | 74.5 | 52.9 | 16.5 | 48.0 |
| **+ DyME** | SigLIP-400M | Qwen2-0.5B | **78.3** (+3.4%) | **67.5** (+6.1%) | **20.4** (+4.5%) | **55.4** (+4.7%) |
| InternVL2-S (Chen et al., 2024) | InternViT-300M | Qwen2-0.5B | 78.3 | 71.9 | 18.7 | 56.3 |
| + Two-stage | InternViT-300M | Qwen2-0.5B | 73.6 | 55.7 | 17.1 | 48.8 |
| **+ DyME** | InternViT-300M | Qwen2-0.5B | **80.0** (+1.7%) | **74.5** (+2.6%) | **19.8** (+1.1%) | **58.1** (+1.8%) |

LVLMs like MoVA (54.2) on these specialized domains, with SmolVLM reaching 55.6 and LLaVA-OV-S 55.4. As a result, `DyME`-trained SVLMs become reliable options for task-specific applications on resource-constrained edge devices.

### 4.2.2 ABLATION STUDY

To dissect the source of these gains, we conducted an ablation study to analyze the contribution of `DyME`'s four core components within the full pipeline: the memorization mode, exploration mode, visual refiner, and visual checker. Table 3 shows the performance impact.

**Dynamic Switching Mechanism.** The results confirm that Memorization and Exploration are symbiotic. Disabling memorization causes a catastrophic drop ($55.4 \rightarrow 43.9$), effectively reverting to unconstrained, unstable exploration. Conversely, removing exploration (50.4) restricts the model to the static imitation of suboptimal data. As shown in Fig. 4, their dynamic interplay prevents the advantage collapse observed in baselines, ensuring optimization stability throughout the learning process.

Table 3: **Ablation study.** Model: LLaVA-OV-S.

| DyME Variant | Medical | Chart | Geometry | Average |
|---|---|---|---|---|
| DyME (full) | 78.3 | 67.5 | 20.4 | 55.4 |
| w/o memorization | 63.2 | 53.4 | 15.0 | 43.9 (20.6%↓) |
| w/o exploration | 75.5 | 61.3 | 14.5 | 50.4 (9.0%↓) |
| w/o visual refiner | 75.6 | 62.3 | 16.8 | 51.6 (6.9%↓) |
| w/o visual checker | 76.9 | 64.3 | 17.1 | 52.8 (4.7%↓) |

**Visual Supervision.** Removing the visual checker and refiner drops performance by $4.7\%$ and $6.9\%$, respectively. This validates the pivotal role of visual supervision in bootstrapping from noisy, undesigned data. Given the limited capacity of SVLMs, they are easily prone to hallucination when trained on low-quality traces. The visual components act as a dynamic denoiser, ensuring that raw, imperfect data is filtered and refined into grounded visual facts ($I_c$) before optimization, thus enabling robust learning even from weak supervision.

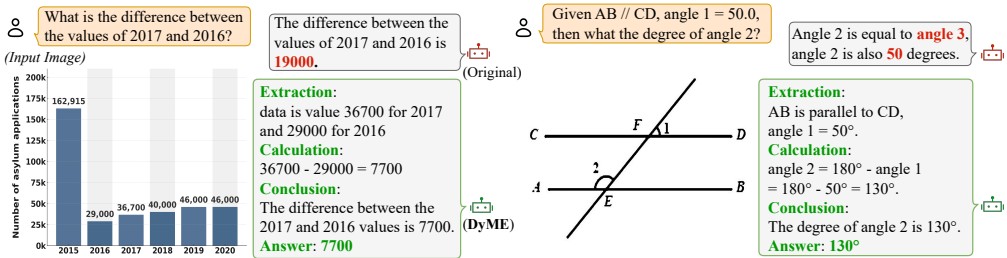

Figure 5: **Showcases on chart understanding and geometry solving.** We use LLaVA-OV-S to demonstrate the results. The SVLM originally produces hallucinated answers (red), while the DyME-trained model generates structured thinking traces (green) that incorporate grounded values, effectively improving the performance.

## 4.3 TRAINING EFFICIENCY & DISCUSSION

We analyze the computational efficiency and performance trade-offs associated with different configurations of `DyME`. The comparative results are detailed in Table 4.

**Computational Efficiency *vs*. Data Cost.** The framework offers two distinct operating regimes catering to different resource profiles. Pure `DyME` represents the high-efficiency regime: when offline CoT data is pre-constructed, it maintains training throughput comparable to standard GRPO (∼14s/step) while delivering superior performance. In contrast, Full `DyME` (with Visual Supervision) prioritizes data autonomy. While the online interaction introduces a computational overhead (∼1.6× training time), it enables the model to bootstrap high-performance reasoning solely from open-source models, bypassing the dependency on expensive, proprietary data annotation (*e.g.*, GPT-4o).

Table 4: **Cost-Benefit Analysis.** Time measured in sec/step. Run on 8x H800.

| Method | Ext. Model | Time | Acc. |
|---|---|---|---|
| GRPO (Baseline) | Qwen2.5-14B[†] | 14.8s | 60.8 |
| Pure `DyME` | Qwen2.5-14B[†] | 14.0s | 64.9 |
| Pure `DyME` | GPT-4o[†] | 19.1s | **68.5** |
| Full `DyME` | Qwen2.5-7B | 21.2s | 66.8 |
| Full `DyME` | Qwen2.5-14B | 23.4s | 67.5 |

[†] Used for offline data construction only.

**Sensitivity to External Model Capacity.** For Full `DyME`, we further examine the impact of the external helper's size on system performance. As shown in Table 4, replacing the Qwen2.5-14B helper with the smaller 7B variant results in a negligible performance variation (67.5% → 66.8%). This indicates that our structured prompt engineering effectively decomposes complex reasoning tasks, allowing even smaller external models to provide sufficient guidance for SVLMs without necessitating heavy-weight models.

**Applicability of Visual Supervision.** The effectiveness of the Visual Supervision module relies on the explicit extraction of Visual Facts ($I_c$). This process creates specific applicability boundaries. For domains involving *abstract semantics* (*e.g.*, irony in memes) or *unstructured perception* (*e.g.*, dense crowds), converting holistic visual signals into discrete text may result in information loss. In such scenarios, reverting to the Pure `DyME` paradigm serves as a more robust alternative.

## 5 CONCLUSION

In this work, we introduced `DyME`, a novel training paradigm designed to empower thinking capabilities within SVLMs. At its core, `DyME` combines memorization (via SFT) mode and exploration (via RLVR) mode through a dynamic switching mechanism. Our experiments demonstrate that this approach not only resolves the critical trade-off between these two modes but also yields substantial performance gains on a wide spectrum of vision tasks, from recognition-intensive to reasoning-intensive scenarios. The success of `DyME` is attributed to its carefully designed components: the dynamic switching mechanism addresses pseudo thinking traces and advantage collapse, while the visual checker and refiner provide coordinated, high-quality visual supervision. It imposes minimal requirements on the base VLM, making it broadly applicable to a wide range of models, including extremely lightweight SVLMs. Therefore, `DyME` serves as the practical solution for empowering SVLMs to think.

ACKNOWLEDGMENT

This work was supported by the Hong Kong SAR RGC General Research Fund (16219025), National Natural Science Foundation of China Young Scholar Fund Category B (62522216), National Natural Science Foundation of China Young Scholar Fund Category C (62402408), and Hong Kong SAR RGC Early Career Scheme (26208924).

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

# Empowering Small VLMs to Think with Dynamic Memorization and Exploration

## Supplementary Material

In the supplementary materials, we report:

- LLM instructions used for constructing vision supervision (§S1);
- Detailed experimental setup and additional experimental results (§S2);
- Showcases of SVLMs trained via DyME performing on medical VQA, chart understanding, and geometry problem solving (§S3);

## S1 LLM INSTRUCTIONS FOR VISION SUPERVISION

The instructions for constructing $I_c$, the visual refiner, and the visual checker are listed as follows.

### S1.1 INSTRUCTIONS FOR EXTRACTING VISUAL ELEMENTS

$I_c$ is primarily derived from two sources: ground truth captions, and the outputs from specialized tools such as the chart-parsing model Deplot. Prompt S1 is employed to extract visual elements from captions.

```
1  You are a helpful assistant that analyzes images and provides visual
       facts.
2  Your response MUST be a single, valid JSON object.
3  The JSON object should contain:
4  1. "description": A detailed and accurate description of the image.
5  2. "objects": A list of key objects, including their name, attributes,
       and approximate position in the image.
6
7  Example format:
8  {
9    "description": "A person riding a bicycle on a city street.... (
         detailed description here)",
10   "objects": [
11     {"name": "person", "attributes": ["wearing helmet", "blue shirt"], "
           position": "center"},
12     {"name": "bicycle", "attributes": ["red", "mountain bike"], "position
           ": "center"},
13     {"name": "street", "attributes": ["asphalt", "wet"], "position": "
           bottom"}
14   ]
15
16 Analyze the attached image and provide the visual facts in the required
       JSON format.
17 For context, the user will be asked this question about the image (do not
        answer the question, just use it for context):
18 "{question}"
19 }
```

Prompt S1: **Automated Visual Fact Extraction**

### S1.2 INSTRUCTIONS FOR VISUAL REFINER

The visual refiner primarily takes thinking templates from a template pool and $I_c$ as input to construct the ground truth for SFT.

```
1  Given:
2  <IC>: the data of an image
3  <Q>: a question
```

```
4   <A>: a reference answer
5   <T>: a writing template
6
7   Goal:
8   Transform the visual information in <IC> into a textualized data
        description and incorporate it into a smooth, natural explanation
        that reasons why the correct answer to <Q> is <A>, using the format
        and tone defined by <T>.
9
10  Example:
11  <IC>: [
12    {"object": "bar", "attributes": ["~120k", "Q4"], "label": "Product A"},
13    {"object": "bar", "attributes": ["~150k", "Q4"], "label": "Product B"},
14    {"object": "bar", "attributes": ["~90k", "Q4"], "label": "Product C"},
15    {"title": "Quarterly Revenue"}
16  ]
17  <Q>: Which product has the highest revenue in Q4?
18  <A>: product b
19  <T>:
20      [Extraction] Extract the relevant data points from <IC> that are
            needed to answer <Q>.
21      [Calculation] Analyze and compare the extracted data to determine the
            correct result.
22      [Conclusion] Based on the analysis, state why <A> is the correct
            answer to <Q>.
23
24  <Output>:
25      [Extraction] The image shows a chart comparing three items-Product A,
            Product B, and Product C-in Q4. The bar heights indicate their
            revenues: Product A at about 120k, Product B at about 150k, and
            Product C at about 90k.
26      [Calculation] Comparing these values, Product B's revenue (150k) is
            greater than both Product A's (120k) and Product C's (90k).
27      [Conclusion] Since Product B has the highest revenue among the three,
            the correct answer to the question "Which product has the
            highest revenue in Q4?" is product b.
28
29  Now, according to the requirements and the examples above, convert my
        input into the target reasoning text. Please give me the result
        directly without any explanation or description.
30
31  <IC>: %s
32  <Q>: %s
33  <A>: %s
34  <T>: %s
35  <Output>:
```

Prompt S2: **Ground-truth construction for Chart Understanding SFT**

Prompts for the other domains follow a similar design.

### S1.3 INSTRUCTIONS FOR VISUAL CHECKER

The visual checker is primarily responsible for scoring the thinking trace of responses generated in the GRPO process. It evaluates these traces with reference to exemplars, based on their fluency and the degree to which the mentioned visual elements align with $I_c$. Prompts for the other domains follow a similar design.

```
1   Given
2   <IC>: the data of an image
3   <Q>: a question
4   <A>: a reference answer
5   <R>: a reasoning text
6
```

```
 7  Goal:
 8  Assess whether <R> correctly and reasonably uses visible data in <IC> to
        justify that the correct answer to <Q> is <A>. Rate the quality as
        low / medium / high according to:
 9  (a) low: Does not use data from <IC> at all, or the language is not
        fluent/natural, or it fails to indicate the answer to <Q> is <A>.
10  (b) medium: Uses data from <IC> and is written fluently, but the
        reasoning is overly brief or insufficiently clear.
11  (c) high: Uses data from <IC> and is written fluently; the reasoning
        progresses step by step with depth, each step is correct and
        reasonable; the data from <IC> appears exactly where it should;
        overall, the reasoning text provides very strong support that the
        answer to <Q> is <A>.
12
13  Example:
14  <IC>: [
15    {"object": "bar", "attributes": ["~120k", "Q4"], "label": "Product A"},
16    {"object": "bar", "attributes": ["~150k", "Q4"], "label": "Product B"},
17    {"object": "bar", "attributes": ["~90k", "Q4"], "label": "Product C"},
18    {"title": "Quarterly Revenue"}
19  ]
20  <Q>: Which product has the highest revenue in Q4?
21  <A>: product b
22  <R>:
23      [Extraction] Reads Q4 bar heights: A ~120k, B ~150k, C ~90k.
24      [Calculation] Compares values: B > A and B > C.
25      [Conclusion] Therefore, Product B is highest, matching the answer "
          product b".
26
27  <Output>: medium
28
29  According to the requirements and examples above, score the input into
        three categories. Please give me the result directly without any
        explanation or description.
30
31  <IC>: %s
32  <Q>: %s
33  <A>: %s
34  <R>: %s
35  <Output>:
```

Prompt S3: **Scoring generations during GRPO for Chart Understanding**

## S2 EXPERIMENTAL DETAILS AND EXTRA RESULTS

### S2.1 TRAINING AND TEST SETTING

First, we provide the statistical information for the training and testing phases of our experiments in the Tab. S1. The training dataset for each domain consists of only a few thousand samples. In addition, Fig. S1 visualizes a comparison between the ground-truth responses produced by the refiner and the original ground-truth, showing that the refined versions are noticeably more structured and place greater emphasis on intermediate values.

Table S1: **Training and testing setup.** DyME empowers thinking capabilities based on small training sets.

| Domain | Training set | #Training samples | Source of $I_c$ | Testset |
|---|---|---|---|---|
| Medical VQA | SLAKE-Train | 4,919 | BiomedGPT | SLAKE-Test |
| Chart Understanding | ChartQA-Train | 4,576 | DePlot | ChartQA-Test |
| Geometry Solving | Geo170K | 6,417 | Collected | MathVerse |

**Medical (SLAKE).**

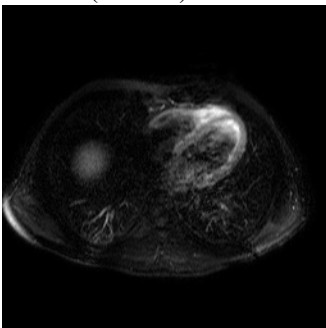

**Q**: What is the MR weighting in this image?

**w/o refiner**:
Based on signal variations typical of this image, the most plausible answer is **T2**.

**w/ refiner**:
*Analysis:* On T2, fluid is bright; bone/air and fast flow are dark.
*Extraction:* In this axial slice, intraluminal fluid is bright; vessels low signal; ribs/wall dark.
*Conclusion:* Pattern matches **T2**.

**Chart (ChartQA).**

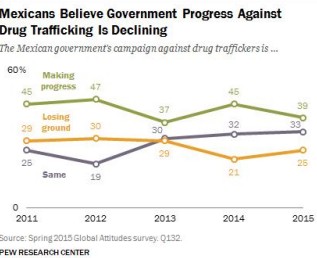

**Q**: What is the sum of *making progress*, *losing ground*, and *same* in 2011?

**w/o refiner**:
To answer the question, we first locate the relevant part of the chart for 2011, then extract the values for the three series and sum them up to obtain **99**.

**w/ refiner**:
*Extraction:* Read 2011 values: Making progress = 45, Losing ground = 29, Same = 25.
*Calculation:* 45 + 29 + 25 = 99.
*Conclusion:* The sum is **99**.

**Math (Geo170k).**

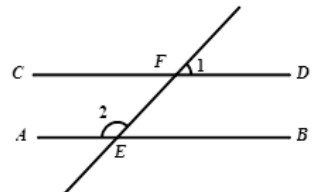

**Q**: If angle 1 is $35°$, what is angle 2? Choices: A: $55°$, B: $135°$, C: $145°$, D: $155°$.

**w/o refiner**:
Since angle 1 is 35 degree, angle 2 is 180 - 35 = 145 degree using a supplementary relationship. Choose **C**.

**w/ refiner**:
*Extraction:* Two parallel lines CD and AB with a transversal; angle 1 is 35 degree, and angle 2 is on the same side.
*Calculation:* Same-side interior angles are supplementary, so angle 2 = 180 - 35 = 145 degree.
*Conclusion:* Answer: **C**.

Figure S1: **Comparison of ground-truth responses before and after refinement.** Compared to the original ground-truth, the refiner injects richer visual elements and enforces a more structured organization, thereby reducing the learning burden for SVLMs.

## S2.2 Extra Results

We also report additional experimental content, including the discussion on training strategies and data organization formats, as well as a comparative analysis with other similar methods that integrate SFT and RL.

Specifically, (1) we first demonstrate the importance of constructing vision supervision, which proves essential for training SVLMs to produce grounded thinking traces. (2) We then examine the impact of structured versus open-ended output formats on thinking performance. (3) Furthermore, to validate our earlier observation that SVLMs are prone to converging to local optima, we present performance across different training epochs, showing that SFT training saturates after only one epoch. (4) We provide a detailed comparison with alternative methods that integrate SFT and RL. (5) Finally, we extend our evaluation to stronger base models and pure textual domains, and (6) validate the quality of generated thinking traces through human evaluation.

Table S2: **Two-stage training on ChartQA**. Rel-corr denotes the relaxed-correctness metric. $I_c$ indicates whether an explicit image-content field is supervised (✓ yes; ✗ no).

| Model | $I_c$ | Rel-corr |
|---|---|---|
| SmolVLM | ✓ | 64.32 |
| SmolVLM | ✗ | 60.09 |
| LLaVA-OV-S | ✓ | 63.62 |
| LLaVA-OV-S | ✗ | 52.90 |

**(1) Intermediate values matter.** As shown in Table S2, we report the effect of applying two-stage training with visual supervision on SmolVLM and LLaVA-OV-S. Incorporating visual supervision significantly improves the best performance achieved during training, despite certain instabilities, thereby validating its critical role for SVLMs. This effect is further illustrated in Fig. S1, where visual supervision compels SVLMs to generate intermediate reasoning enriched with visual elements, which make a clear contribution to the final answer.

**(2) Structured thinking alleviates the learning burden of SVLMs.** Table S3 reports the performance gap between training with structured thinking ground-truth and with unconstrained ground-

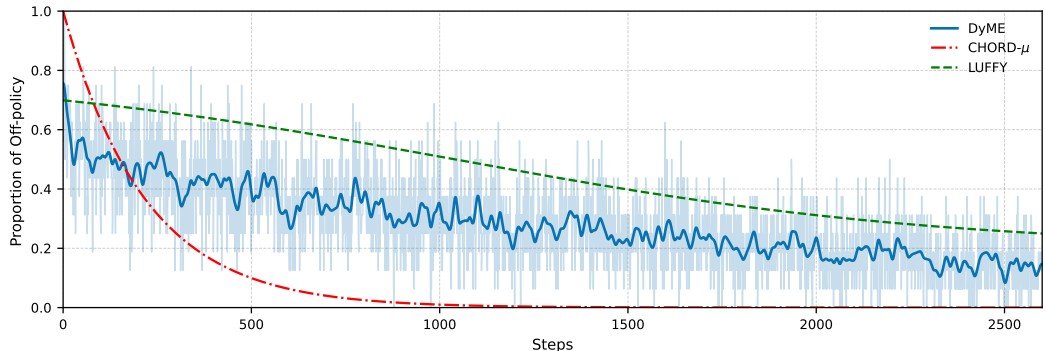

Figure S2: **Relative off-policy influence during training.** Each curve is normalized to its initial value for comparability. DyME measures $\mathrm{SFT}/(\mathrm{SFT} + \mathrm{RL})$ (raw in lighter tone, Gaussian-smoothed in darker tone), CHORD-$\mu$ tracks the global weight $\mu(t)$, and LUFFY adopts a policy-shaping proxy $\mathbb{E}[f(\pi_\theta(a))]$ with $f(x) = \frac{x}{x+\gamma}$. All methods reveal the shift from off-policy guidance to on-policy optimization, albeit with distinct decay dynamics.

Table S3: **Effect of templated output across models and tasks.** ✓ denotes fixed-template output, whereas ✗ denotes free-form generation.

| Model | Template | Chart | Medical |
|---|---|---|---|
| SmolVLM | ✓ | 60.10 | 59.38 |
| SmolVLM | ✗ | 59.24 | 56.13 |
| LLaVA-OV-S | ✓ | 52.87 | 74.52 |
| LLaVA-OV-S | ✗ | 50.86 | 72.64 |

truth. While open-ended exploration is often beneficial for LVLMs, the limited capacity of SVLMs makes unconstrained exploration less effective, as it tends to be aimless and increases the learning burden. Given that SVLMs are designed for task-specific rather than general-purpose scenarios, employing tailored thinking templates for each task proves more suitable and yields better performance. For instance, SmolVLM achieves $60.10$ *vs.* $59.24$ on ChartQA and $59.38$ *vs.* $56.13$ on Medical VQA, with LLaVA-OV-S exhibiting similar gains.

**(3) Comparison between annealed SFT loss and DyME.**
As shown in Fig. S2, we compare the relative SFT (off-policy) influence across training steps for three approaches: DyME, CHORD (Zhang et al., 2025b), and LUFFY (Yan et al., 2025). For DyME and CHORD, the curves represent the normalized weight of the SFT loss at each step, while for LUFFY the curve reflects the trajectory of SFT gradient shaping as a function of prediction probability (which generally correlates with training steps). These curves highlight the dynamic nature of DyME. Because of the extremely limited capacity of SVLMs, their learning patterns can shift significantly even between adjacent steps, leading to rapid forgetting of previously acquired modes. Unlike CHORD, which relies on a smooth annealing schedule that decays quickly and is ill-suited to such small models, DyME assigns weights directly based on model outputs.

This produces a highly dynamic and irregular decay, better accommodating the instability of SVLMs. LUFFY adopts a shaping function $f(x) = \frac{x}{x+\gamma}$ ($\gamma$=0.1), which also induces a dynamic decay with probability but remains heuristic and may not align well with the local-optimum tendency of SVLMs. Overall, DyME is explicitly tailored for SVLMs, whereas

Table S4: **SVLM performance saturates after a single training epoch.** Score is domain-specific: chart domain uses Rel-corr, while the medical domain uses the average of accuracy and recall values.

| Model | Domain | Epoch | Score |
|---|---|---|---|
| LLaVA-OV-S | Chart | 1 | 60.70 |
| | | 5 | 60.44 |
| | | 10 | 60.12 |
| SmolVLM | Chart | 1 | 60.22 |
| | | 5 | 63.21 |
| | | 10 | 62.22 |
| | Medical | 1 | 71.73 |
| | | 5 | 71.80 |
| | | 10 | 72.05 |

Table S5: **Detailed learning trajectories demonstrating rigorous tuning.** We report the performance across multiple settings to show their full learning trajectories. Two-stage baselines include variants with and without KL penalties to ensure optimal performance is captured.

| Data Quality | Method | Performance across epochs (1, 3, 5, 10) | Best perf. |
|---|---|---|---|
| Low | DyME (ours, pure) | *Report final score directly* | **61.9** |
| | SFT | $43.1 \rightarrow 47.9 \rightarrow 50.0 \rightarrow 50.5$ | 50.5 |
| | Two-stage | $57.6 \rightarrow 52.7 \rightarrow 50.8 \rightarrow 50.7$ | 57.6 |
| | Two-stage (w/ KL) | $54.2 \rightarrow 55.4 \rightarrow 52.6 \rightarrow 54.2$ | 55.4 |
| Medium | DyME (ours, pure) | *Report final score directly* | **64.9** |
| | SFT | $53.6 \rightarrow 56.5 \rightarrow 57.8 \rightarrow 56.4$ | 57.8 |
| | Two-stage | $59.9 \rightarrow 52.8 \rightarrow 53.0 \rightarrow 53.1$ | 59.9 |
| | Two-stage (w/ KL) | $59.0 \rightarrow 60.6 \rightarrow 60.6 \rightarrow 60.8$ | 60.8 |
| High | DyME (ours, pure) | *Report final score directly* | **68.5** |
| | SFT | $58.2 \rightarrow 59.1 \rightarrow 61.0 \rightarrow 61.6$ | 61.6 |
| | Two-stage | $51.6 \rightarrow 54.0 \rightarrow 54.5 \rightarrow 54.4$ | 54.5 |
| | Two-stage (w/ KL) | $61.7 \rightarrow 60.9 \rightarrow 62.7 \rightarrow 61.8$ | 62.7 |

CHORD and LUFFY may be more appropriate for stronger base models, reflecting complementary strengths.

**(4) SVLMs converge rapidly.** Table S4 shows that SVLMs converge extremely quickly: performance after only one epoch is comparable to, or even exceeds, that after ten epochs (*e.g.*, LLaVA-OV-S achieves 60.70 *vs.* 60.12 on the Chart domain). This indicates that the very limited capacity of SVLMs makes them prone to overfitting to local optima. It also substantiates our earlier claim that such rapid convergence leaves only a narrow window for balancing SFT and RL, making it difficult to achieve the trade-off through empirical hyperparameter tuning. Consequently, static fusion methods are unsuitable for SVLMs.

To ensure a rigorous comparison, we further report the full learning trajectories of baselines in Table S5. We evaluated the Two-stage baseline (with and without KL penalty) and SFT across multiple epochs (1, 3, 5, 10) to capture their peak performance. The results confirm that even with optimal stopping, the baselines consistently underperform DyME, which achieves superior results in a single training run without the need for epoch selection.

**(5) Generality across complex reasoning and pure text.** To demonstrate the scalability of DyME, we applied it to two new domains without modifying the core algorithm: Physical Reasoning (A-OKVQA) and **Pure Text Reasoning** (GSM8K).

- **Physical Reasoning (A-OKVQA):** We addressed the challenge of open-ended visual reasoning by testing on A-OKVQA. We used Qwen2.5-VL-7B to automatically generate Visual Facts using the prompt defined in §S1 (e.g., *"man, wearing a light blue and white shirt..."*). As shown in Table S6, DyME achieved a massive gain of +18.8% ($54.2\% \rightarrow 73.0\%$), proving that the method scales effortlessly to tasks requiring world knowledge and commonsense.

- **Pure Text Reasoning (GSM8K):** In pure text domains, the "Visual Fact" extraction step is naturally skipped. On the GSM8K math benchmark, DyME improved Qwen2.5-0.5B from $49.5\%$ to $55.3\%$, demonstrating that the paradigm generalizes even when "vision" is absent.

These results, combined with the ChartQA improvements on the stronger Qwen2.5-VL-7B model, confirm that DyME is not limited by the extraction step. By leveraging off-the-shelf LVLMs to automate visual fact generation, the framework is immediately applicable to diverse visual and textual domains.

**Limitations on Abstract Visuals.** We acknowledge that the VS module may face challenges in scenarios where "Visual Facts" are intrinsically difficult to define or extract, such as memes (relying on irony or cultural context) or highly abstract non-commonsense reasoning. However,

Table S6: **Generality of `DyME` across New Domains.** We demonstrate performance gains on Complex Scenes (A-OKVQA), Pure Text (GSM8K), and with stronger base models (Qwen2.5-VL-7B). Baselines for text use standard GRPO.

| Domain | Task | Base Model | Method | Baseline (%) | `DyME` (%) |
|---|---|---|---|---|---|
| **World Knowledge** | A-OKVQA | LLaVA-OV-S | Two-stage | 54.2 | **73.0 (+18.8)** |
| **Pure Text** | GSM8K | Qwen2.5-0.5B | GRPO | 49.5 | **55.3 (+5.8)** |
| **New LVLM** | ChartQA | Qwen2.5-VL-7B | SFT | 87.3 | **89.6 (+2.3)** |

our primary objective is to empower SVLMs for practical, real-world production tasks (*e.g.*, chart processing, medical diagnostics, geometric solving). In these structured and semi-structured domains where SVLMs are most commonly deployed, Visual Facts are well-defined and `DyME` proves highly effective.

**(6) Human evaluation of CoT quality.** Automatic metrics like relaxed accuracy do not fully reflect the quality of the reasoning process. To verify whether `DyME` generates genuinely better thinking traces, we conducted a human evaluation on 100 randomly sampled instances from ChartQA. Annotators judged the validity of the generated CoT based on its logical coherence and grounding. As shown in Table S7, `DyME` produces traces that are slightly more concise (shorter length) but significantly more valid (validity rate ~70%) compared to the Two-stage baseline (~30-40%). This confirms that `DyME` effectively mitigates the generation of "pseudo thinking traces" that plague standard SFT/Two-stage training.

Table S7: **Human evaluation of CoT quality on ChartQA.**

| Base Model | Method | Avg. CoT Length | Human Eval (Valid %) |
|---|---|---|---|
| **LLaVA-OV-S** | Two-stage | ~76.3 Words | 31% |
|  | DyME | ~69.7 Words | **68%** |
| **SmolVLM** | Two-stage | ~84.5 Words | 40% |
|  | DyME | ~75.4 Words | **72%** |

## S3 SHOWCASES

Before presenting the model outputs, we first illustrate the data quality definitions used in our Algorithmic Validation (Section 4.1 of the main paper). Table S8 showcases examples of Low (Undesigned), Medium (Standard), and High (Premium) quality Chain-of-Thought supervision for the same question. This visualizes the significant gap in structure and detail that `DyME` must bridge when trained on non-premium data. Furthermore, Table S9 illustrates the comprehensive format of our supervision data, encompassing the input image, the associated question, the extracted visual facts, and the ground-truth response.

We present dialogue instances of SmolVLM, LLaVA-OV-S, and InternVL2-S, which were trained with `DyME` in our experiments, on tasks in the domains of medical VQA, chart understanding, and geometry. As shown in Fig S3 to Fig S5, all models trained with `DyME` demonstrate the ability to generate effective thinking traces with accurate intermediate values (in green), which play a crucial role in reaching the final correct answer.

The thinking processes of these models are relatively fixed and template-based. This is a result of our vision supervision module taking effect, as SVLMs should not be overly broad and general; otherwise, their exploration can easily diverge instead of converging.

Table S8: **Showcase of chain-of-thought (CoT) supervision with different quality for ChartQA.**

| Type | Content |
| --- | --- |
| Question | When does the unfavorable view reach the peak?  |
| High-quality CoT | <SUMMARY> I inspect the line graph showing very unfavorable views of Pakistan in India over time, and identify the year where the percentage peaks by locating the maximum point on the curve. </SUMMARY> |
| | <CAPTION> Line graph titled "Very unfavorable views of Pakistan increasing in India". The y-axis shows the share of respondents with a very unfavorable view (0–100%). The x-axis shows years 2013–2017. Marked values: 54% (2013), 49% (2014), 51% (2015), 55% (2016), and 64% (2017). The series exhibits an overall upward trend and peaks in 2017. </CAPTION> |
| | <REASONING> The peak corresponds to the highest percentage on the y-axis. Among the annotated years, 2017 has the largest value (64%), higher than 2013–2016. Therefore, the peak occurs in 2017. </REASONING> |
| | <CONCLUSION> 2017 </CONCLUSION> |
| Medium-quality CoT | **Goal:** Determine the year when the unfavorable view reaches its maximum. **Observation:** The values for each year are: 2013: 0, 2014: 0, 2015: 0, 2016: 55, and 2017: 64. **Reasoning:** Comparing these values, the largest value is 64, which appears in 2017. **Conclusion:** The unfavorable view reaches its peak in 2017. |
| Low-quality CoT | I'm trying to figure out the year when the unfavorable view reaches its highest point. The values stay low until 2016, where they rise to 55, and then climb further to 64 in 2017. This makes 2017 the year when the unfavorable view reaches its maximum. |

Table S9: **Showcase of a complete CoT training example.**

| Type | Content |
| --- | --- |
| Question | What is the man by the bags awaiting?  |
| Visual Fact | { "description": "A man stands in the middle of a residential street, facing away from the camera. He holds a red bag and pulls a wheeled black suitcase, with another black suitcase placed nearby. Houses, parked cars, and trees line the background, suggesting he is waiting to cross or be picked up.", "objects": [ "name": "man", "attributes": ["light blue and white shirt", "blue jeans", "carrying a red bag", "pulling a wheeled suitcase"], "position": "center" , "name": "red bag", "attributes": ["held by the man"], "position": "left of the man" , "name": "black suitcase", "attributes": ["wheeled", "being pulled"], "position": "near the man's feet" , "name": "black suitcase", "attributes": ["lying on the ground"], "position": "near the man" , "name": "street", "attributes": ["asphalt", "urban residential"], "position": "foreground" , "name": "houses", "attributes": ["residential homes"], "position": "background left" , "name": "parked cars", "attributes": ["red SUV", "other vehicles"], "position": "background center and left" , "name": "trees", "attributes": ["green foliage"], "position": "background right" ] } |
| GT Response | A train would not appear on a residential street, and the man's luggage suggests he is waiting for transportation rather than a delivery. The nearby skateboarder is not interacting with him, leaving a cab as the most plausible option. |

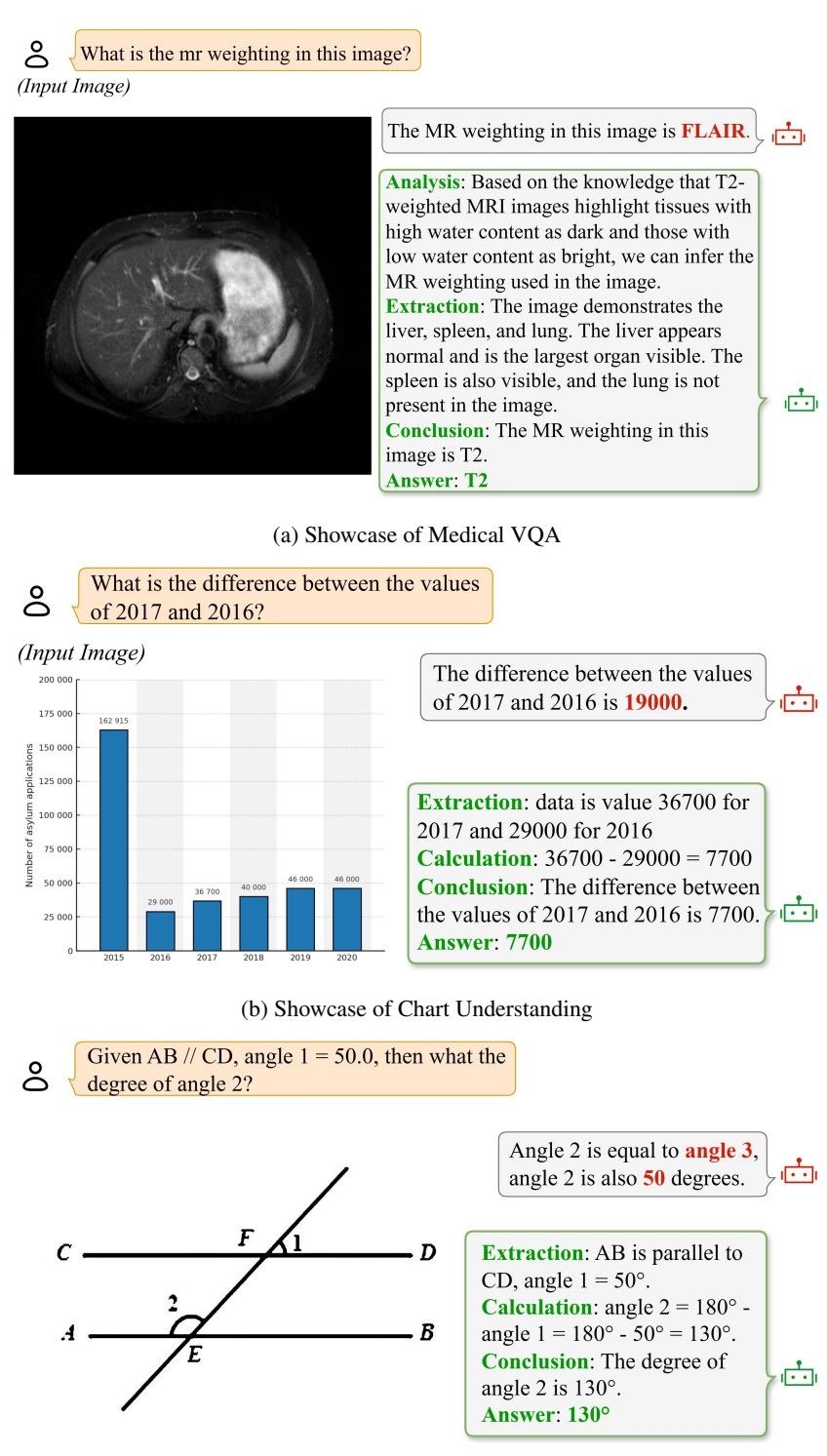

(a) Showcase of Medical VQA

(b) Showcase of Chart Understanding

(c) Showcase of Geometry Solving

Figure S3: **Showcases of SmolVLM.** The SVLM originally produces hallucinated answers (red), while the `DyME`-trained model generates structured thinking traces (green) that incorporate grounded values, effectively improving the performance.

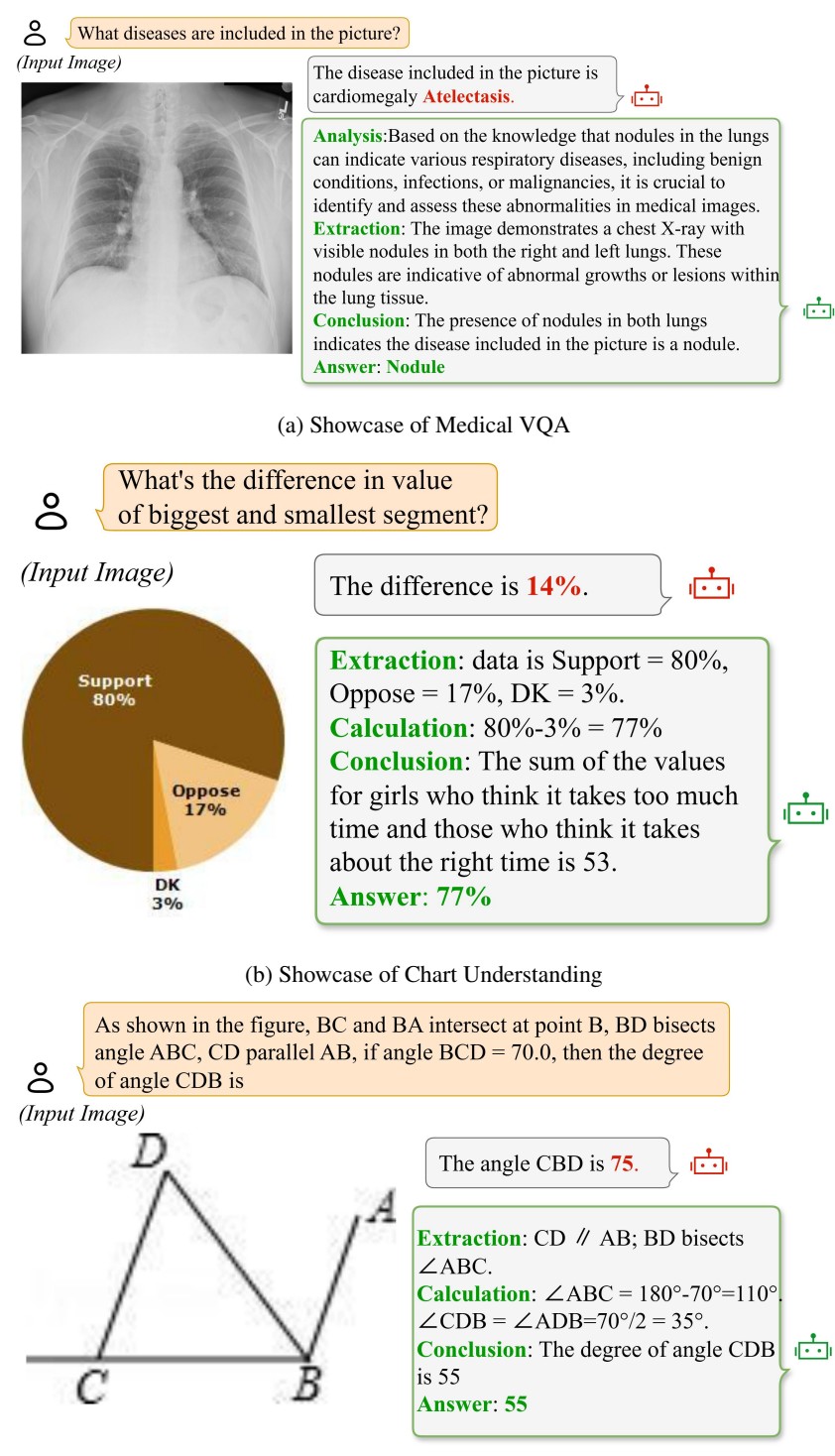

(a) Showcase of Medical VQA

(b) Showcase of Chart Understanding

(c) Showcase of Geometry Solving

Figure S4: **Showcases of InternVL2-S.** The SVLM originally produces hallucinated answers (red), while the `DyME`-trained model generates structured thinking traces (green) that incorporate grounded values, effectively improving the performance.

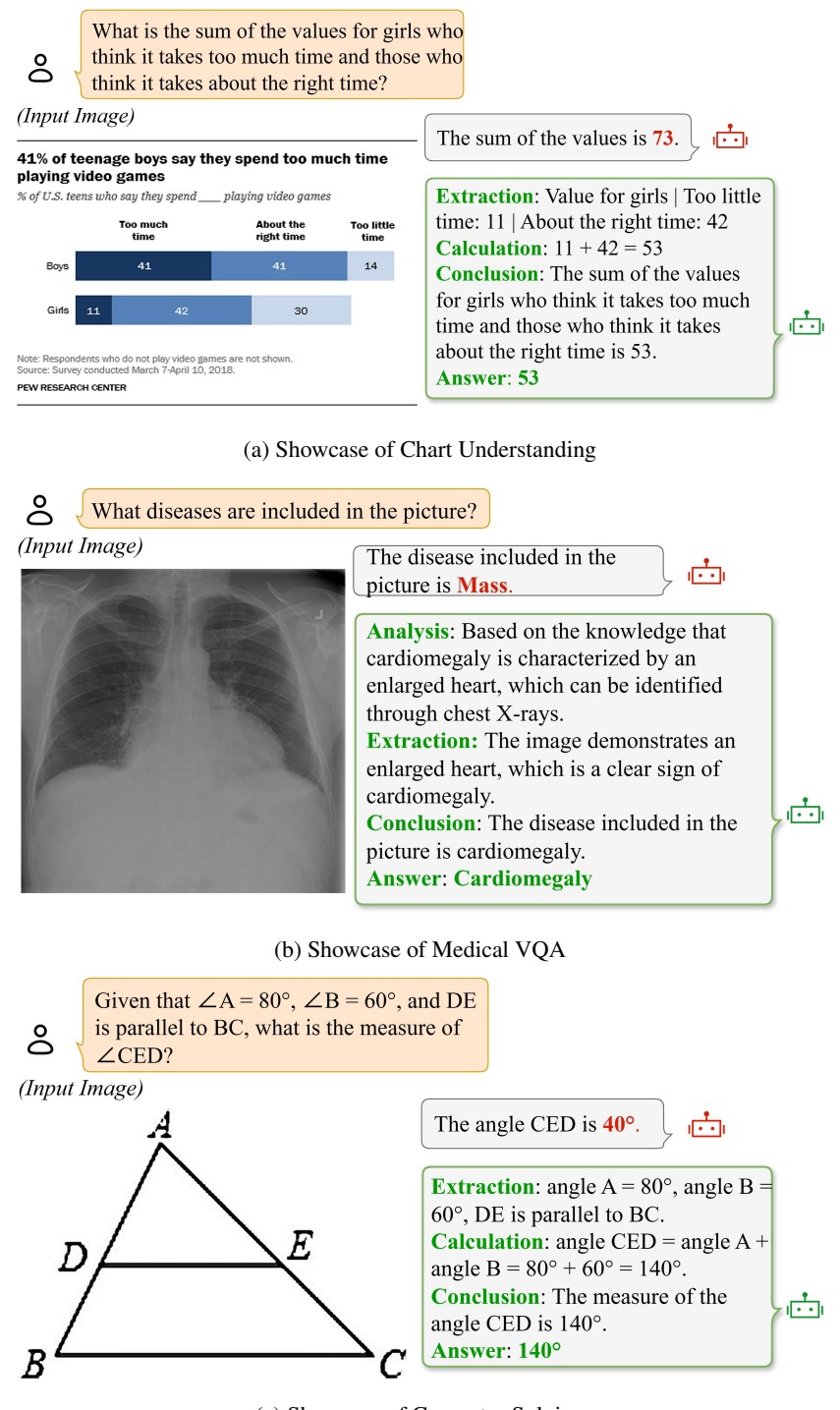

(a) Showcase of Chart Understanding

(b) Showcase of Medical VQA

(c) Showcase of Geometry Solving

Figure S5: **Showcases of LLaVA-OV-S.** The SVLM originally produces hallucinated answers (red), while the `DyME`-trained model generates structured thinking traces (green) that incorporate grounded values, effectively improving the performance.

