# OpenReview forum: "Empowering Small VLMs to Think with Dynamic Memorization and Exploration"
_ICLR.cc/2026/Conference — ICLR 2026 Poster_

### Official Review · Reviewer_mKuR · 2025-11-01

**Soundness:** 2
**Presentation:** 3
**Contribution:** 2
**Rating:** 2
**Confidence:** 3

**Summary:**

This paper tackles a critical and practical challenge: enabling reasoning (“thinking”) in Small-scale Vision-Language Models (SVLMs). The authors argue that existing training paradigms for Large VLMs—Supervised Fine-Tuning (SFT) on CoT data and Reinforcement Learning with Verifiable Reward (RLVR)—are ill-suited for SVLMs. SFT overwhelms small models, harming visual grounding, while RLVR often collapses due to poor instruction-following and unstable training.
To address this, the authors propose DyME (Dynamic Memorize–Explore), a novel training paradigm that dynamically switches between SFT and RLVR at each optimization step:
- Dynamic Switching: If at least one of the multiple generated responses is correct, DyME enters Exploration mode (RLVR) to encourage diverse reasoning. If all fails, it reverts to Memorization mode (SFT) to learn from ground-truth traces.
- Complementary Supervision: A visual checker rewards visually grounded reasoning, while a visual refiner enhances SFT targets using successful exploration traces.
Experiments on three domains—medical VQA, chart understanding, and geometry reasoning—show that DyME yields substantial and consistent gains across multiple SVLMs, often matching or surpassing larger models.

**Strengths:**

1. Timely and Meaningful Problem: Focuses on reasoning for small, efficient VLMs—highly relevant for real-world deployment on edge devices.
2. Elegant and Effective Approach: The dynamic “memorize–explore” mechanism intuitively balances stability and exploration, well-suited to SVLM limitations.
3. Strong Experimental Evidence:
  - Baselines clearly show SFT/RLVR failures, motivating DyME.
  - Consistent, significant improvements across all domains.
  - Ablation studies confirm each component’s necessity and synergy.
4. Excellent Clarity: The paper is clearly written, with strong visuals (notably Fig. 1) and a logical, accessible presentation.

**Weaknesses:**

1. Reliance on External LLM: The visual checker and refiner rely on a large external model (Qwen2.5-14B), introducing extra complexity, cost, and dependency. This makes performance partly contingent on the external LLM’s capability, slightly undermining the goal of a self-contained small-model framework.
2. Rigid Switching Heuristic: The binary rule (“if one correct → RLVR, else → SFT”) may cause abrupt shifts; a softer, reward-based switch could yield smoother training.
3. Limited Task Generality: The method requires domain-specific Visual Fact extraction, which may hinder scalability to new, open-ended tasks.

**Questions:**

1. External LLM Sensitivity: How would performance change if smaller or open-source models were used for the visual checker/refiner?
2. Training Overhead: What is the computational and time cost of DyME relative to standard SFT and RLVR?
3. Effect of K: How does the number of generated responses per step (K) affect stability, performance, and cost?
4. Visual Fact Extraction: For novel tasks not covered in the paper (e.g., complex scene understanding  or physical reasoning), what is the anticipated process for extracting the Visual Facts? Does this step require significant manual design, or can it be automated?

**Details Of Ethics Concerns:**

no concerns.

---

> ### Author Response · Authors · 2025-11-20
> **[1/3] Response to Reviewer mKuR**
>
> We sincerely thank you for recognizing our approach as “elegant and effective” and for appreciating the timeliness of the problem. To fully address your concerns regarding external dependencies, switching heuristics, and task scalability, we have conducted extensive new experiments.
>
> *   **Reproducibility:** We have uploaded our full codebase, training scripts, and processed data to the anonymous repository link [here](https://anonymous.4open.science/r/2276-rebuttal-F215/).
>
> ### **1. Addressing “Reliance on External LLMs” & “Self-Contained” Goal (Weakness 1 & Q1)**
>
> We address this concern with a strong conclusion: **DyME achieves state-of-the-art performance even in a strictly self-contained setup (without the external VS module).**
>
> **1.1 Strong Proof: Pure DyME Outperforms Baselines.**
> To prove this, we evaluated **"Pure DyME"** (where the Online VS module is deactivated). In this setting, the training process is **fully self-contained**, using the **exact same static offline data** as the baselines.
>
> *   **Setup:** We constructed static CoT datasets for ChartQA using Qwen2.5-14B (Medium Quality) and GPT-4o (High Quality). We then trained LLaVA-OV-S on these identical datasets using SFT, Two-stage, and Pure DyME.
> *   **Results (Table R3-1):** Pure DyME consistently outperforms SFT and Two-Stage baselines. Notably, using only Medium-quality data, Pure DyME (64.9%) surpasses the Two-Stage baseline trained on premium High-quality data (62.7%).
>
> **Table R3-1: Performance comparison using identical Offline Teacher Data.**
> Pure DyME operates without any online external loop, using the exact same static data as baselines.
>
> | Data Source (Teacher)                 | Method (Student) | Performance (%) |
> |:--------------------------------------| :--- |:---------------:|
> | **Offline: Medium Quality**           | SFT |      57.8       |
> | *(Source: Qwen2.5-14B)*               | Two-stage |      60.8       |
> |                                       | **Pure DyME (Ours)** | **64.9 (+4.1)** |
> | **Offline: High Quality**             | SFT |      61.6       |
> | *(Source: GPT-4o)*                    | Two-stage |      62.7       |
> |                                       | **Pure DyME (Ours)** | **68.5 (+5.8)** |
> | **Online: DyME + Visual Supervision** | |                 |
> | *(Source: Qwen2.5-14B)*               | **DyME (Full)** |    **67.5**     |
>
> **Key Takeaways from Controlled Experiments:**
> 1.  **DyME is a “Better Student,” Independent of External Modules:** Pure DyME outperforms baselines by a large margin (+4.1% to +5.8%) on identical static data. This proves our gains stem from the robustness of the DyME algorithm itself, not just the external module.
> 2.  **Democratizing Reasoning Training:** By comparing the results, we find that DyME + VS (using open-source Qwen2.5-14B) achieves 67.5%, effectively matching the performance of expensive, proprietary GPT-4o data (68.5%). Thus, instead of relying on closed-source APIs like typical paradigms, DyME enables high-capability training using **accessible open-source compute**, making the framework truly practical.
>
> **1.2 The “External Dependency” Stems from SVLMs, Not DyME.**
>
> With the empirical proof established, we respectfully clarify the theoretical root of external dependence: **The need for external knowledge is a fundamental prerequisite for SVLMs, not a specific limitation of our algorithm.**
>
> As detailed in recent technical reports (e.g., *SmolVLM* from HuggingFace team), pre-trained SVLMs intentionally exclude heavy CoT data to preserve visual capabilities. Consequently, off-the-shelf SVLMs possess no intrinsic reasoning capability to self-discover. Thus, all paradigms must inject external knowledge:
>
> *   **Baselines** rely on an **Offline Teacher** (static data from external LLMs).
> *   **DyME** relies on the same **Offline Teacher** in its Pure mode, but can optionally leverage an **Online Teacher** (VS) for better results.
>
> **Conclusion:** DyME does not create a new dependency; it simply offers a **better student algorithm** (the switching mechanism) that learns more effectively from *any* teacher, whether offline (Pure) or online (Full).
>
> ### **2. Sensitivity to External Models & Training Overhead (Weakness 1, Q1 & Q2)**
> Based on the Pure DyME findings above, we frame the Visual Supervision module not as a limitation, but as a **flexible strategic choice** that allows users to trade training compute time for massive savings in data acquisition costs.
>
> **2.1 Cost-Benefit Trade-off (Answer to Q2).**
>
> We quantified the training overhead in Table R3-2. The choice depends on the user's resource constraints:
> *   **Efficiency-Oriented (Pure DyME):** If high-quality offline data is already available (or throughput is the priority), Pure DyME is the optimal choice. By removing the external model, it eliminates online overhead, achieving training speeds comparable to the GRPO baseline (14.0s vs. 14.8s per step) while still delivering superior results (+4.1% gain).

---

> > ### Author Response · Authors · 2025-11-20
> > **[2/3] Response to Reviewer mKuR**
> >
> > *   **Cost-Oriented (DyME + VS):** If high-quality data is unavailable or too expensive (e.g., no budget for GPT-4o), DyME + VS provides an automated solution. Although it incurs an online inference overhead (~9s latency), it effectively bootstraps GPT-4o level performance using only open-source models, bridging the data quality gap without the premium price tag.
> >
> >
> > **2.2 Robustness to Model Size (Answer to Q1).**
> >
> > For users who opt for the VS module, we further validated that **performance is not sensitive to the external model's size**. As shown in Table R3-2, replacing Qwen2.5-14B with the smaller Qwen2.5-7B results in negligible performance loss (67.5% $\rightarrow$ 66.8%). This is because our prompt engineering decomposes complex reasoning into simple sub-tasks manageable by smaller models.
> >
> > **Table R3-2: Cost-Benefit Analysis of Offline vs. Online Supervision.** Training time represents the average duration per step (in seconds), measured under identical settings (TRL framework, 8x H800 GPUs, ZeRO-2, Batch Size=4, K=4). The external model for GRPO and pure DyME is only used during data construction, not training.
> >
> > | Category       | External Model Source | Performance (%) | Training Time (s/global step) |
> > |:---------------|:----------------------|:---------------:|:-----------------------------:|
> > | **GRPO**       | Qwen2.5-14B        |      60.8       |             14.8              |
> > | **Pure DyME**  | Qwen2.5-14B           |      64.9       |             14.0              |
> > | **Pure DyME**  | GPT-4o                |    **68.5**     |             19.1              |
> > |
> > | **DyME + VS**  | Qwen2.5-7B            |      66.8       |             21.2              |
> > | **DyME + VS**  | Qwen2.5-14B           |      67.5       |             23.4              |
> > |  **DyME + VS**              | Qwen2.5-32B           |      67.6       |             34.9              |
> >
> > ### **3. Addressing the Rigid Switching Heuristic (Weakness 2)**
> >
> > You raised an insightful point suggesting that our binary switching rule might cause “abrupt shifts” and that a “softer, reward-based switch” could yield smoother training.
> >
> > We respectfully demonstrate that the binary switch is neither mathematically abrupt nor empirically inferior. In fact, it proves to be the optimal design for SVLMs.
> >
> > **3.1 Theoretical Justification: The Switch is Mathematically Smooth.**
> >
> > We explicitly address the concern of “abrupt shifts” through the **Gradient Compatibility** analysis in Section 3.1 (Eq. 4 & 5) of our paper.
> > *   We proved that the SFT gradient is formally a special case of the GRPO gradient (where the ground-truth sample is assigned unit advantage).
> > *   Therefore, switching between modes does **not** alter the fundamental optimization landscape or cause conflicting gradient updates. It merely shifts the *source* of the advantage signal (from self-generated exploration to ground-truth guidance). This theoretical alignment ensures that the binary switch is seamless rather than disruptive.
> >
> > **3.2 Empirical Validation: Binary Outperforms “Soft Strategies”.**
> >
> > To empirically validate this, we compared our Binary Switch against the suggested Soft Reward Threshold and other hybrid strategies (Table R3-3). The alternatives are:
> > 1.  **Reward Threshold:** Switch to RL only if batch avg. reward $> t$ (Soft Switch).
> > 2.  **SFT Annealing:** Always add SFT loss, decaying its weight over time (Hybrid Loss).
> >
> >
> > **Table R3-3: Ablation of switching strategies (ChartQA, Pure DyME of LLaVA-OV-S).**
> >
> > | Method | Hyperparameter | Perf. (%) | Computational Overhead |
> > | :--- | :--- | :---: | :---: |
> > | **DyME (Ours)** | **Binary Switch** | **64.9** | **Baseline** |
> > | Reward Threshold | $t=0.5$ | 52.4 | No |
> > | | $t=0.8$ | 64.1 | No |
> > | | $t=0.9$ | 63.4 | No |
> > | SFT Annealing | Cosine Schedule | 64.0 | +25% |
> >
> > **Key Findings:**
> > 1.  **Softer is More Brittle:** The *Reward Threshold* method is highly sensitive to the hyperparameter $t$. A suboptimal threshold ($t=0.5$) caused performance to collapse (52.4%). Even with exhaustive tuning ($t=0.8$), it failed to surpass the parameter-free DyME (64.9%).
> > 2.  **Binary is Robust:** The binary rule effectively acts as a dynamic safety gate. It ensures the model *only* explores when it has proven capability (at least one success) and *immediately* stabilizes with SFT when it fails, preventing the “aimless exploration” typical of SVLMs.
> > 3.  **Hybrid Methods Tax Efficiency:** *SFT Annealing* achieves decent performance (64.0%) but imposes a +25% computational tax at every step.  This tax arises because the design adds a mandatory SFT computation to the K rollouts required for each RL update (effectively 5 forward/backward passes vs. 4).
> >
> > **Conclusion:** The binary switch is not a rigid heuristic but an **empirically optimal and theoretically sound design** that balances stability and exploration without requiring brittle hyperparameter tuning.

---

> > > ### Author Response · Authors · 2025-11-20
> > > **[3/3] Response to Reviewer mKuR**
> > >
> > > ### **4. Scalability to New Domains & Automated Extraction (Weakness 3 & Q4)**
> > >
> > > You expressed concern that “domain-specific Visual Fact extraction” might hinder scalability. We clarify this by distinguishing between the **DyME algorithm** and the **VS module**:
> > >
> > > **4.1. DyME (The Algorithm) is Universal.**
> > >
> > > The DyME paradigm itself does **not** limit generalization. It acts as a universal, superior alternative to standard baselines (SFT/RL) in *any* scenario.
> > > *   As proven in Part 1, without online Visual Supervision (i.e., **Pure DyME**), our switching mechanism alone stabilizes training and outperforms baselines.
> > > *   Therefore, DyME is applicable anywhere baselines are applicable, regardless of whether visual extraction is possible.
> > >
> > > **4.2. VS (The Module) is Broadly Applicable.**
> > > The question then becomes: *How limited is the optional VS module?* We argue that it is highly scalable because **Visual Fact extraction is fully automated** via off-the-shelf LVLMs.
> > > *   **Automated Extraction:** We use a generic prompt (*"Describe objects and attributes..."*) to extract facts using Qwen2.5-VL.
> > > *   **New Experiments:** To address your specific example, we applied this pipeline to **AOKVQA** (requiring world knowledge and commonsense reasoning). DyME achieved a massive gain of **+18.8%** (54.2% $\rightarrow$ 73.0%), demonstrating the strong generality of DyME in handling open-ended visual reasoning tasks.  The result is in Table R3-4.
> > >
> > > **Table R3-4: Generality of DyME across New Domains.**
> > >
> > > | Domain | Task | Base Model | Baseline (%) | **DyME (%)** |
> > > |:---|:---|:---| :---: | :---: |
> > > | **World Knowledge** | AOKVQA | LLaVA-OV-S | 54.2 | **73.0** |
> > >
> > > **4.3. Handling Corner Cases (Abstract Concepts).**
> > > We acknowledge that VS may face challenges in scenarios where Visual Facts are intrinsically undefined, such as memes (relying on irony/cultural context) or highly abstract art.
> > > *   **In these rare cases:** Users can simply skip the extraction step and use **Pure DyME**.
> > > *   **Conclusion:** Since Pure DyME is already strictly better than standard baselines, our framework remains the optimal choice even when the VS module is not used.
> > >
> > >
> > > ### **5. Effect of Sample Size $K$ (Answer to Q3)**
> > >
> > > We empirically tested $K \in \{4, 8, 16\}$ using Pure DyME. Performance plateaued or even slightly dropped: **64.9%** ($K=4$), **63.7%** ($K=8$), and **64.6%** ($K=16$).
> > > *   **Analysis:** Unlike LVLMs where increasing $K$ improves coverage of diverse reasoning paths, SVLMs exhibit exploration saturation. Due to limited capacity, the valid reasoning space is narrower; simply increasing sampling width ($K$) tends to capture more hallucinated noise rather than novel correct solutions.
> > > *   **Efficiency:** Since training cost rises linearly with $K$ without yielding performance gains, we selected **$K=4$** as the optimal balance of stability and efficiency.

---

### Official Review · Reviewer_6oCC · 2025-11-01

**Soundness:** 3
**Presentation:** 3
**Contribution:** 2
**Rating:** 4
**Confidence:** 4

**Summary:**

This paper addresses the challenge of instilling reasoning ("thinking") capabilities in SVLMs. SFT overwhelms the models' limited capacity, leading to "pseudo thinking traces," while RLHR fails due to poor instruction adherence, causing "advantage collapse."

Thus, the authors propose DyME (Dynamic Memorize-Explore), a novel training paradigm. DyME dynamically switches between an exploration (RLVR) mode and a memorization (SFT) mode at each optimization step. The switch is governed by a simple rule: if the model fails to produce any correct response in a batch, it falls back to SFT mode. If at least one response is correct, it uses RLVR mode.

Furthermore, DyME introduces a "vision supervision" mechanism, composed of a "Visual Checker" and "Visual Refiner", which use an external LLM to (1) provide a more nuanced "thinking reward" for the RLVR mode and (2) dynamically refine the SFT ground-truth targets to be more visually grounded and consistent with successful exploration traces.

Experiments across three domains show that DyME substantially improves the performance of SVLMs, whereas the SFT and RLVR baselines are shown to degrade performance.

**Strengths:**

1. The paper tackles a practical and important problem of enabling complex reasoning on small, efficient models.

2. The paper is generally well-written, and the problem is clearly motivated. Figure 1, in particular, provides a good illustration of why existing paradigms might fail on SVLMs and how DyME tries to solve this.

3. Strong empirical results. The authors show in Tab. 1 that standard SFT, RLVR, and a two-stage approach degrade the performance of SVLMs on these tasks, while DyME consistently provides significant gains.

4. Effective ablation study. The ablation in Table 2 does a good job of validating the key components of DyME. It shows that removing either the memorization or exploration mode is catastrophic, supporting the need for a hybrid approach. It also demonstrates the effectiveness of the visual supervision modules.

**Weaknesses:**

1. The "vision supervision" modules (Visual Checker and Refiner) involves additional dependencies. The modules are critical to the method's performance (as shown in Tab. 2), but they are implemented via prompting an external Qwen2.5-14B. This involves additional knowledges and causes unfair comparisons.

2. Question regarding novelty of this work. Compared with existing hybrid SFT+RL methods, the authors claim the main novelty is the dynamic switching criterion. However, the criterion used (fall back to SFT if all $K$ samples are incorrect) is a relatively simple and heuristic. The paper note that prior hybrid methods are "static," but this might be an oversimplification. A deeper comparison to other dynamic weighting schemes (e.g., PPO-SFT hybrids) may be helpful to fully position the novelty of this specific heuristic.

3. The fact that all baselines (SFT, GRPO, Two-stage) fail so catastrophically (e.g., dropping average performance from 49.9 to 44.1 or 44.0 for SmolVLM) is surprising. While I understand the  limited capacity of SVLMs can lead to underperforming results, I wonder if this could be caused by under-optimized tuning. For example, SFT's failure is attributed to overwhelming the modell, but could a simpler baseline (e.g., SFT on far less data or shorter CoT) have been more effective? I would appreciate any deeper discussion into this concern.

4. Clarity of "Vision Supervision" and lack of details. While the ablation shows the visual checker/refiner are important, their description in the main paper is high-level. More detail on the prompts, the failure modes of this LLM-based pipeline, and its reliability would be
necessary to make the method truly reproducible.

5. The visual refiner/checker requires additional inference of a LVLM during the training of a LVLM. This seems time consuming and may harm the effectiveness of the proposed training pipeline. Could you quantify the computational cost of this pipeline? How critical is the choice of Qwen2.5-14B? What happens if a weaker/stronger model is used for the checker and refiner? Does the method still work?

6. More investigation into the switching criterion. The binary switch (all fail vs. $\ge 1$ success) is simple and effective. Did you experiment with other criteria (e.g., switching if the average reward of the batch is below a threshold, or using a "budget" for SFT steps) that might be more robust?

7. It seems the authors omit the visual checker/refiner in the abstract. To fully reflect the contribution of this paper, they may consider adding this part into the text.

**Questions:**

Please see the comments above regarding the weaknesses. I have written how each concern can be discussed and addressed in the rebuttal/revision.

---

> ### Author Response · Authors · 2025-11-20
> **[1/3] Response to Reviewer 6oCC**
>
> We thank you for recognizing our strong empirical results and the problem's significance. To address your concerns regarding comparison fairness, baseline optimization, and cost, we have conducted extensive new experiments and released our code to fully resolve these issues.
>
> *   **Reproducibility**: The datasets, training logs, and full codebase for these experiments are available in our anonymous repository [here](https://anonymous.4open.science/r/2276-rebuttal-F215/) for verification.
>
> ### **1. Concern on Visual Supervision (VS) & Fairness**
>
> We thank you for raising this point. To fully resolve the fairness concern regarding the Qwen2.5-14B dependency, we explicitly decouple the algorithmic contribution of DyME from the data enhancement provided by the VS module.
>
> **1.1 Decoupling Algorithm from Data Source.**
>
> Fundamentally, all training paradigms for SVLMs require external supervision to instill reasoning capabilities, as these small models cannot self-discover reasoning through zero-shot exploration. The distinction lies only in the format: baselines rely on static **offline** CoT datasets, while our full DyME utilizes dynamic **online** supervision via VS.
>
> To ensure a strictly fair comparison, we strip away the VS module and its external dependency. This yields **Pure DyME**, which uses the exact same static CoT data as the baselines, isolating the impact of our dynamic training paradigm.
>
> **1.2 Experiments on Pure DyME.**
>
> We conducted controlled experiments on ChartQA using LLaVA-OV-S, where SFT, Two-Stage, and Pure DyME were trained on identical CoT datasets of varying qualities (Low/Medium/High), with the Visual Supervision (VS) module explicitly excluded.
> *   **Setup:** The training data consists of standard ChartQA query-answer pairs augmented with CoT traces pre-constructed by external LLMs. Visual samples are available in the anonymous repository.
> *   **Robust Baselines**: To rule out suboptimal tuning, we performed an exhaustive sweep for all baselines across epochs and hyperparameters. Table R2-1 reports the single best performance achieved for each baseline, while Table R2-2 details the learning trajectories to demonstrate the tuning process.
>
> **Table R2-1: Performance comparison on LLaVA-OV-S (ChartQA) using IDENTICAL CoT datasets.** Note that for baselines, we report the best performance achieved via exhaustive hyperparameter grid search.
>
> | Data Quality | Description | Method               | Best Perf. (%)  |
> | :--- | :--- |:---------------------| :---: |
> | **Low** | *Undesigned* (~80 words; unstructured; noisy & verbose) | SFT                  | 50.5 |
> | | | Two-stage            | 57.6 |
> | | | **Pure DyME (Ours)** | **61.9** |
> | **Medium** | *Semi-designed* (~89 words; semi-structured; partial visual grounding; **via Qwen2.5-14B**) | SFT                  | 57.8 |
> | | | Two-stage            | 60.8 |
> | | | **Pure DyME (Ours)** | **64.9** |
> | **High** | *Fine-designed* (~142 words; highly Structured; rich visual grounding; quality-controlled; **via GPT-4o**) | SFT                  | 61.6 |
> | | | Two-stage            | 62.7 |
> | | | **Pure DyME (Ours)** | **68.5** |
>
>
> **Result**: As shown in Table R2-1, Pure DyME consistently outperforms all baselines across all data splits, proving that **our gains stem from the robust Memorize-Explore switching mechanism, not just the external model**:
> 1.  **Robustness to Noise:** On Low-quality data, where SFT struggles (50.5%) and Two-Stage is unstable due to noise (57.6%), Pure DyME effectively filters noise and stabilizes learning, achieving 61.9%.
> 2.  **Higher Ceiling:** Even with top-tier data (GPT-4o), DyME (68.5%) still beats the best-tuned baseline (62.7%) by +5.8%. This proves Pure DyME utilizes high-quality data more efficiently than standard paradigms.
>
> **Crucial Insight: VS Bridging the Data Quality Gap.** By cross-referencing Table R2-1 with our main paper results (Revised version, Table 2), a powerful conclusion emerges: DyME + VS (using open-source Qwen2.5-14B) achieves 67.5%, effectively matching the performance of Pure DyME trained on expensive GPT-4o data (68.5%). This validates that the VS module acts as a high-quality data synthesizer, allowing users to achieve GPT-4o level performance with open-source costs.

---

> > ### Author Response · Authors · 2025-11-20
> > **[2/3] Response to Reviewer 6oCC**
> >
> > ### **2. Catastrophic Failure of Baselines & Tuning**
> > To determine whether the catastrophic failure is merely an artifact of under-optimization, we conducted a comprehensive sweep across varying data qualities, CoT lengths, training epochs, and hyperparameters (Table R2-2). Our findings reveal that **the failure is not an artifact of under-optimization, but rather a demonstration of the inherent fragility of standard paradigms on SVLMs.**
> >
> > **Table R2-2: Detailed learning trajectories demonstrating rigorous tuning.** We report the performance across multiple settings to show their full learning trajectories.
> >
> > | Data Quality | Method | Performance across Epochs (1, 3, 5, 10) | Best Perf. |
> > | :--- | :--- |:----------------------------------------| :---: |
> > | **Low** | **DyME (ours, pure)** | *Report final score directly*     | **61.9** |
> > | | SFT | 43.1 → 47.9 → 50.0 → 50.5               | 50.5 |
> > | | Two-stage | 57.6 → 52.7 → 50.8 → 50.7               | 57.6 |
> > | | Two-stage (w/ KL) | 54.2 → 55.4 → 52.6 → 54.2               | 55.4 |
> > | **Medium** | **DyME (ours, pure)** | *Report final score directly*     | **64.9** |
> > | | SFT | 53.6 → 56.5 → 57.8 → 56.4               | 57.8 |
> > | | Two-stage | 59.9 → 52.8 → 53.0 → 53.1               | 59.9 |
> > | | Two-stage (w/ KL) | 59.0 → 60.6 → 60.6 → 60.8               | 60.8 |
> > | **High** | **DyME (ours, pure)** | *Report final score directly*     | **68.5** |
> > | | SFT | 58.2 → 59.1 → 61.0 → 61.6               | 61.6 |
> > | | Two-stage | 51.6 → 54.0 → 54.5 → 54.4               | 54.5 |
> > | | Two-stage (w/ KL) | 61.7 → 60.9 → 62.7 → 61.8               | 62.7 |
> >
> > We acknowledge that under idealized conditions, baselines can avoid collapse. For instance, *only* by combining High Quality data, KL penalties, and precise early stopping (Epoch 5), does the Two-stage baseline reach 62.7%. However, even this peak performance significantly lags behind Pure DyME (68.5%) and relies on unrealistically perfect tuning. The results highlight that baselines suffer from:
> >
> > *   **Sensitivity to Hyperparameters:** As shown in Table R2-2, the impact of the KL penalty is inconsistent. On High quality data, adding KL improves the Two-stage baseline significantly (54.5% $\rightarrow$ 62.7%). However, on Low quality data, adding KL actually hurts performance (57.6% $\rightarrow$ 55.4%). Also, SFT on Medium data improves initially but degrades after epoch 5 (57.8% $\rightarrow$ 56.4%), requiring precise early stopping. This implies that standard baselines require a grid search for every new dataset.
> > *   **Structure > Length:** Simply shortening CoT does not solve the bottleneck (e.g., the performance on 80-word unstructured CoT is significantly lower than on 142-word structured data). SVLMs cannot afford to waste limited capacity on meaningless connecting text or unstructured diversity.
> >
> > Thus, the catastrophic failure observed in our main paper is not an artifact of sub-optimization, but rather reflects the inherent instability of these paradigms. **The Success Window for Baselines is Extremely Narrow.** This precisely validates our motivation: to design DyME as a robust paradigm that guarantees training stability.
> >
> >
> > ### **3. On Novelty: Why the Binary Switch is Optimal**
> >
> > We thank you for suggesting these alternative switching criteria. We conducted specific ablations on the Medium dataset for Pure DyME (where it achieves 64.9%) to validate our design.
> >
> > **3.1 Comparison with Alternative Strategies.**
> > We compared DyME against three suggested dynamic strategies:
> > 1.  **Reward Threshold:** Switch to RL only if batch avg. reward $> t$ (Soft Switch).
> > 2.  **SFT Annealing:** Always add SFT loss, decaying its weight over time (Hybrid Loss).
> > 3.  **SFT Budget:** Accumulate all-wrong samples into a buffer and perform focused SFT updates (Hard Negative Mining).
> >
> > The results are shown in Table R2-3.
> >
> > **Table R2-3: Ablation of switching strategies (ChartQA-Medium, Pure DyME).**
> >
> > | Method | Hyperparameter | Perf. (%) | Computational Overhead |
> > | :--- | :--- | :---: | :---: |
> > | **DyME (Ours)** | **Binary Switch** | **64.9** | **Baseline** |
> > | Reward Threshold | $t=0.5$ | 52.4 | No |
> > | | $t=0.8$ | 64.1 | No |
> > | | $t=0.9$ | 63.4 | No |
> > | SFT Annealing | Cosine Schedule | 64.0 | +25% |
> > | SFT Budget | Budget Size = `3 * N_nodes * N_devices` | 59.6 | Based on budget |
> >
> > **3.2 Key Findings:**
> > 1.  **Softer is More Brittle:** The *Reward Threshold* method is highly sensitive. A suboptimal threshold ($t=0.5$) degrades performance to 52.4%. Finding the magic number ($t=0.8$) requires exhaustive tuning for every setting, whereas DyME's binary rule is parameter-free and robust.
> > 2.  **Hybrid Methods Tax Efficiency:** *SFT Annealing* (64.0%) incurs a +25% computational overhead by enforcing a mandatory SFT gradient calculation alongside the $K$ (=4) rollouts. This is strategically inefficient, as it indiscriminately applies SFT to every sample, even trivial ones the model has already mastered.

---

> > > ### Author Response · Authors · 2025-11-20
> > > **[3/3] Response to Reviewer 6oCC**
> > >
> > > 3. **Hard Negative Mining Destabilizes SVLMs:** The *SFT Budget* approach performs poorly (59.6%). Overwhelming a low-capacity SVLM with a concentrated batch of its own failures leads to instability rather than improvement.
> > >
> > > **Conclusion:** Our binary switch is not merely a heuristic; it is the empirically optimal design for SVLMs, balancing performance, robustness, and efficiency better than complex alternatives.
> > >
> > > ### **4. Computational Cost & Sensitivity to External Models**
> > > We acknowledge the valid concern regarding the additional inference cost and dependency introduced by the VS module. To quantify this impact and assess robustness, we performed the following analyses (Table R2-4).
> > >
> > > **4.1 Robustness to Model Size.**
> > >
> > > We explicitly tested replacing the Qwen2.5-14B in the VS module with the smaller Qwen2.5-7B. The performance drop is negligible (67.5% $\rightarrow$ 66.8%), confirming that our prompt engineering successfully decomposes complex reasoning into simple sub-tasks manageable by smaller models.
> > >
> > > **4.2 Cost-Benefit Trade-off.**
> > >
> > > While VS introduces online overhead, we frame this not as a limitation, but as a flexible strategic choice. Users can choose where to allocate resources, either in upfront data engineering or online computation:
> > > *   **Efficiency-Oriented (Pure DyME):** If users invest effort or budget beforehand to pre-construct high-quality data (e.g., via GPT-4o), Pure DyME is the optimal choice. It removes the external loop, achieving maximum training throughput (~14.0s/step) comparable to baselines, yet consistently outperforms them under these identical data conditions (e.g., +4.1% over Two-stage on Qwen2.5-14B data, see Table R2-1).
> > > *   **Cost-Oriented (DyME + VS):** If high-quality data construction is too tedious or expensive, DyME + VS offers an automated solution. It shifts the burden to dynamic, open-source computation. Although this incurs online overhead, it allows the model to reach the same high-performance ceiling without heavy upfront data costs. Notably, Table R2-4 shows that DyME + VS (using Qwen2.5-14B) achieves 67.5%, effectively matching Pure DyME trained on premium GPT-4o data (68.5%).
> > >
> > >
> > > **Table R2-4: Cost-Benefit Analysis of Offline vs. Online Supervision.** Training time represents the average duration per step (in seconds), measured under identical settings (TRL framework, 8x H800 GPUs, ZeRO-2, Batch Size=4, K=4). The external model for GRPO and pure DyME is only used during data construction, not training.
> > >
> > > | Category       | External Model Source | Performance (%) | Training Time (s/global step) |
> > > |:---------------|:----------------------|:---------------:|:-----------------------------:|
> > > | **GRPO**       | Qwen2.5-14B        |      60.8       |             14.8              |
> > > | **Pure DyME**  | Qwen2.5-14B           |      64.9       |             14.0              |
> > > | **Pure DyME**  | GPT-4o                |    **68.5**     |             19.1              |
> > > | **DyME + VS**  | Qwen2.5-7B        |      66.8       |             21.2              |
> > > | **DyME + VS**  | Qwen2.5-14B     |      67.5       |             23.4              |
> > > |  **DyME + VS** | Qwen2.5-32B    |      67.6       |             34.9              |
> > >
> > > ### **5. Clarity & Abstract Updates**
> > >
> > > **5.1 Implementation & Failure Modes of VS module.**
> > >
> > > We have provided full prompts in the Supplementary Material and released the code in our anonymous repository.
> > >
> > > **Failure Modes:** We will include a discussion on the limitations of the VS module. Since VS relies on the explicit extraction of Visual Facts ($I_c$), it faces challenges in extraction bottlenecks:
> > >
> > > - For images involving abstract semantics (e.g., emotions, irony in memes) or highly unstructured perception (e.g., dense crowds, low-level textures), extracting discrete, structured visual facts is often infeasible or results in significant information loss.
> > >
> > > In scenarios where extraction is infeasible, the framework seamlessly reverts to Pure DyME (by disabling VS), a setting which, as we have demonstrated, still significantly outperforms standard baselines.
> > >
> > > **5.2 Reliability.**
> > > The pipeline is highly robust to the choice of external models. As shown in Table R2-4, replacing Qwen2.5-14B with the smaller 7B version results in negligible performance loss (67.5% $\rightarrow$ 66.8%). This confirms that DyME is effective even with lightweight external support. Our prompt engineering (see in the codebase) significantly lowers the capability threshold, enabling even lightweight models to generate high-quality supervision.
> > >
> > > **5.3 Abstract Update.**
> > > We agree with your suggestion. We will revise the abstract to explicitly decouple the two contributions:
> > > 1.  **DyME (The Paradigm):** We will emphasize that this alone serves as a robust, standalone strategy that stabilizes SVLM learning (as proven by our Pure DyME results).
> > > 2.  **VS (The Module):** We will clarify that this is a distinct module that synergizes with the core paradigm to maximize performance.

---

### Official Review · Reviewer_y5pj · 2025-11-01

**Soundness:** 3
**Presentation:** 3
**Contribution:** 3
**Rating:** 6
**Confidence:** 4

**Summary:**

The paper proposes a new dynamic training paradigm that can switch between the supervised finetuning and the group relative policy optimization for a vision-language model. Specifically, the paper first  asks the small vision language model to generate the responses. When at least one is correct, the model will choose GRPO, otherwise it will use the SFT-based model. For GRPO reward, apart from the binary correctness, the paper also introduces a visual checker to evaluate the image grounding. The proposed method is evaluated on SLAKE, ChartQA, and MathVerse. The paper uses several VLM baselines to show its performance. The experiment results show that the small model can achieve comparable or even better performance on multiple tasks, when comparing with LVLMs. The experiment also includes ablation study,

**Strengths:**

1. The proposed new training procedures seem to be simple yet effective. DyME can be applied to SVLM and can achieve significant performance gains across different domains. The method can also reduce the advantage collapse and constrained exploration.
2. The experiment is comprehensive. The paper compares the proposed training strategy with two-stage, GRPO, and SFT on three different models with 0.5-1B parameters. The ablation study shows the importance of the proposed training strategy and visual rewards.
3. The paper provides additional training details and examples in the appendix. The illustrative figures help readers to understand the paper better.

**Weaknesses:**

1. Some baselines are pretty old. The paper needs to include some newer LVLMs such as QWen-2.5VL, etc. The experiment section is also purely a quantitative evaluation. Some qualitative evaluation or human evaluation can help readers to understand the quality of the chain better. For example, the length of COT after using DyME compared to the two-stage. Adding additional experiments, such as pure textual or pure vision tasks, can help readers understand the performance gain better. The current evaluation focus on the VQA tasks, which are a bit limited.
2. Some parts of the paper are not clearly written. For example, why use Geo170k for training but evaluate on MathVerse? The paper also mentioned the chartqa used relaxed correctness. What is the approximation used for the evaluation?
3. The paper fails to show any code and model, making it hard for readers to reproduce results. The paper did not include reproducibility statement. The paper fails to include a use of LLMs section.

**Questions:**

What is the performance gain of SVLM on pure text tasks?

---

> ### Author Response · Authors · 2025-11-20
> **Response to Reviewer y5pj**
>
> We sincerely thank you for your positive assessment and for recognizing DyME as a “simple yet effective” method with “comprehensive” experiments. Your suggestions regarding newer baselines and qualitative evaluations were incredibly valuable.
>
> In response, we have conducted new experiments on Qwen2.5-VL and GSM8K, performed a human evaluation of reasoning traces, and released our code. We believe these additions fully address your concerns.
>
> ## **1. Newer and Larger Models & Pure Text Tasks**
>
> **1.1 DyME Scales to Stronger, Newer LVLMs.**
> We applied DyME to the state-of-the-art **Qwen2.5-VL-7B**. As shown in Table R1-1, DyME successfully improves this strong baseline on ChartQA from 87.3% to 89.6%. This confirms that DyME is not limited to SVLMs but is a strong paradigm that may further refine powerful, recent models.
>
> **1.2 Generalization to Pure-Text Reasoning.**
> To test if DyME benefits pure textual reasoning, we evaluated it on the **GSM8K** benchmark using **Qwen2.5-0.5B**. DyME achieved a significant gain (+5.8%) over the GRPO baseline (reported in *SimpleRL-Zoo, COLM 2025*). This demonstrates that our dynamic switching mechanism effectively balances exploration and memorization in non-visual domains as well.
>
> **Table R1-1: Performance of DyME on a new LVLM and in the textual domain.**
>
> | Domain | Task | Base Model | Baseline (%) | **DyME (%)** |
> |:---|:---|:---|:---:|:---:|
> | **New LVLM** | ChartQA | Qwen2.5-VL-7B | 87.3 | **89.6 (+2.3)** |
> | **Pure Text** | GSM8K | Qwen2.5-0.5B | 49.5 | **55.3 (+5.8)** |
>
> ### **2. Qualitative & Human Evaluation**
> You specifically asked about the quality of the chain and the length of CoT after using DyME compared to the two-stage.
>
> **2.1 Human Evaluation.**
> We conducted a blind human evaluation on 100 randomly sampled outputs from ChartQA. Three graduate students rated each reasoning trace as valid only if it was both logically correct and visually grounded (majority vote).
>
> **2.2 Result.**
> As shown in **Table R1-2**, DyME produces reasoning traces that are shorter but significantly more valid compared to the Two-Stage baseline.
> *   The Two-Stage baseline often generates pseudo-thinking traces, i.e., long, repetitive, or hallucinated text that mimics the style of CoT but fails to ground visual facts (hence the lower validity and longer length).
> *   **DyME:** By dynamically maintaining the training balance, DyME curbs these hallucinations, resulting in concise, effective reasoning.
>
> **Table R1-2: Human Evaluation of CoT Quality on ChartQA.**
>
> | Base Model | Method | Avg. CoT Length | Human Eval (Valid %) |
> |:---|:---|:---:|:---:|
> | **LLaVA-OV-S** | Two-stage | ~76.3 Words | 31% |
> | | **DyME** | **~69.7 Words** | **68%** |
> | **SmolVLM** | Two-stage | ~84.5 Words | 40% |
> | | **DyME** | **~75.4 Words** | **72%** |
>
> ### **3. Clarifications**
> *   **Geo170k (Train) $\rightarrow$ MathVerse (Test):** The primary reason is that Geo170K serves strictly as a training dataset and does not provide a standardized test benchmark. Therefore, we utilized MathVerse as the testbed to quantitatively evaluate the model's geometric reasoning capabilities. This setup is consistent with established protocols in prior works like *MoVA (NeurIPS 2024)*.
> *   **Relaxed Correctness (ChartQA):** We followed the standard evaluation script from *DePlot (ACL 2023)* and *ChartQA (ACL 2022)*. Relaxed correctness allows a 5% tolerance for numerical answers (e.g., if the ground truth is 100, an answer of 96-104 is accepted). This is standard practice to account for minor visual ambiguities in reading charts.
>
> ### **4. Reproducibility & LLM Policy**
> *   **Code Release:** We have uploaded our full codebase, training scripts, and processed data to the anonymous repository link [here](https://anonymous.4open.science/r/2276-rebuttal-F215/).
> *   **LLM Policy:** In the revised paper, we will add a dedicated `Use of LLMs` section explicitly detailing that Qwen2.5-14B was used strictly for the *Visual Refiner* and *Visual Checker* modules during training (and for prompt engineering), ensuring full transparency and compliance.

---

> ### Comment · Reviewer_y5pj · 2025-11-26
>
> Thank you for your detailed clarifications! I decide to raise my score to 8.

---

> > ### Author Response · Authors · 2025-11-27
> >
> > Thank you very much for your constructive feedback and for kindly raising your score. We will incorporate the suggested changes in the final manuscript and are grateful for your help in strengthening the paper.

---

### Author Response · Authors · 2025-11-20
**Common Response to All Reviewers**

We sincerely thank all reviewers for their thoughtful and constructive feedback. We are encouraged that you found the problem important and the proposed DyME paradigm simple yet effective, and we appreciate the concrete suggestions on fairness, external dependencies, novelty, and qualitative evaluation.

In the revised manuscript and additional experiments, we have made the following main changes:

1. **Clarified core contribution.** We now clearly separate **DyME (the dynamic Memorize–Explore paradigm)** from the **Visual Supervision (VS) module**, and explicitly position VS as an *optional* enhancement rather than a required component of DyME.

2. **Fairer and stronger baselines.** We added **Pure DyME** experiments that remove VS and train on the **same offline CoT data** as SFT/two-stage. With extensive hyperparameter and epoch sweeps, Pure DyME consistently outperforms all baselines across Low/Medium/High data quality, addressing concerns about fairness and the possibility that “catastrophic failure” is due to insufficient tuning.

3. **Broader evaluation.** We extended our study beyond the original SVLM setting by

   * applying DyME to a **stronger LVLM** (Qwen2.5-VL-7B), and
   * evaluating DyME on a **pure-text reasoning benchmark (GSM8K)**,
     showing that the dynamic switching mechanism remains beneficial across models and domains.

4. **Quality and cost analyses.** We added (i) a **human evaluation of CoT traces**, confirming that DyME produces shorter but more valid and visually grounded reasoning than two-stage training, and (ii) a **detailed cost/sensitivity analysis** of the VS module with respect to external model size and training time, clarifying the trade-off between Pure DyME and DyME+VS.

5. **Reproducibility and transparency.** We added a dedicated **LLM usage statement**, discussed limitations of VS, and released our **full anonymized code, processed data, prompts, and logs** via the anonymous repository linked in the rebuttal.

Please refer to our individual responses for detailed discussions on specific points; we have addressed each weakness and question raised by every reviewer point-by-point.

---

### Author Response · Authors · 2025-12-01
**Summary of Rebuttal Revisions and General Response**

Dear Reviewers and ACs,

We sincerely thank the reviewers for their constructive feedback. We are encouraged that Reviewer y5pj raised their score to **8 (Accept)** following our rebuttal, praising the new experiments and clarifications. While the discussion period was interrupted before we received an initial response from Reviewers 6oCC and mKuR, we have provided comprehensive responses and new experiments to address their concerns regarding **fairness, external dependencies, and generalization**.

Below is a summary of the key revisions and experimental additions included in the updated PDF:

**1. Decoupling the Algorithm from Data Enhancement (“Pure DyME”)**

*(Addressing concerns from R-6oCC and R-mKuR regarding fairness and external dependency)*

A primary concern was whether our performance gains stemmed from the core DyME algorithm or the external Visual Supervision (VS) module (Qwen2.5-14B). To address this, we introduced “Pure DyME”—a strictly self-contained version of our method that removes the online VS module and uses the exact same static CoT data as the baselines.

- Result: Pure DyME consistently outperforms SFT and Two-Stage baselines (+4.1% to +5.8%) on identical data.
- Conclusion: This proves that our dynamic switching mechanism is the core driver of stability and performance, while the VS module acts as an optional, high-quality data synthesizer.

**2. Rigorous Tuning & Rebuttal of Under-optimization**

*(Addressing concerns from R-6oCC regarding baseline failure)*

We addressed the suspicion that the baseline failures were due to under-optimization by conducting an exhaustive sweep of hyperparameters and training epochs (Table R2-2).

- Results: Standard paradigms on SVLMs exhibit an extremely narrow success window (e.g., performance peaks briefly then degrades rapidly due to overfitting or advantage collapse).  Even under perfect tuning conditions, the strongest baseline (Two-stage w/ KL) still significantly underperforms Pure DyME (62.7% vs. 68.5% on High-quality data).
- Conclusion: This confirms the failure is inherent to the paradigm's instability on small models, not a lack of tuning.  Solving this critical stability issue is precisely the primary motivation and contribution of DyME.

**3. Generalization to SOTA Models, Pure Text, and General Scenes**

*(Addressing concerns from R-y5pj and R-mKuR regarding scope and model size)*

We extended our evaluation beyond SVLMs to demonstrate that DyME is a universal training paradigm:
*   Stronger Base Models: Applied DyME to Qwen2.5-VL-7B, improving ChartQA performance to 89.6% (+2.3%), showing DyME refines even strong, state-of-the-art models.
*   Pure Text Domain: Applied DyME to GSM8K (Math) using Qwen2.5-0.5B, achieving +5.8% over the GRPO baseline.
*   World Knowledge: Applied DyME to A-OKVQA (General Scenes), achieving a massive +18.8% gain. Crucially, this validates that the Visual Supervision extraction is fully automated via generalist VLMs (e.g., Qwen2.5-VL) using generic prompts, proving the framework is highly scalable and not limited by domain-specific engineering.

**4. Justification of the Binary Switching Mechanism**

*(Addressing concerns from R-6oCC and R-mKuR regarding heuristics)*

We conducted a rigorous ablation study comparing our binary switch against softer alternatives (Reward Thresholding, SFT Annealing, Hard Mining).
- Results: Soft thresholds are brittle and require per-dataset tuning, while annealing incurs a heavy computational tax (+25%).
- Conclusion: The binary switch is empirically optimal for SVLMs; it acts as a robust safety gate that immediately stabilizes training when exploration fails due to limited model capacity.

We hope that these revisions and clarifications fully address the reviewers’ concerns and further strengthen the overall narrative of DyME as a practical, robust, and effective paradigm for equipping small-scale VLMs with thinking capabilities. All revisions and new results are **highlighted in BLUE** in the updated submission, and all experiments and code have **been released** for full transparency and verification.

---

### Meta-Review · Area_Chair_2ej2 · 2026-01-12

**Summary:**

Reviewers generally agreed the paper tackles an important and timely problem: how to reliably train small VLMs to produce useful reasoning traces without collapsing (RL) or overfitting into pseudo-thinking (SFT).

The main concerns were:
- whether gains are really from the DyME switching paradigm vs. the external VS module / external teacher strength
- whether baseline failures were due to under-tuning
- the computational overhead and sensitivity to the external model choice
- whether the binary switching rule is a brittle heuristic

**Reviewer Concerns:**

Addressed:

-Fairness or external dependency: new **Pure DyME** setting removes the online VS loop and shows consistent gains over SFT/two-stage under identical CoT data, addressing the concern that improvements come **just** from a strong external model.

- Baseline under-optimization: authors report extensive sweeps and learning trajectories; the strongest tuned baselines still underperform Pure DyME, which supports the claim that instability is not merely a tuning artifact.

- Switching heuristic: Added theoretical justification + empirical ablations versus softer/hybrid alternatives; binary switch appears robust and parameter-free relative to reward-threshold tuning and annealing overhead.

- Scope and qualitative validation: additional experiments on newer/stronger models and new domains (Qwen2.5-VL-7B, GSM8K, A-OKVQA) plus a human eval showing improved CoT validity/grounding.

- Revised paper includes an LLM usage statement and a reproducibility/code release link.


Partially outstanding:

- Generality of the VS pipeline / visual fact extraction for highly unstructured or abstract visual reasoning

- Compute/engineering overhead remains a practical drawback for the full DyME+VS variant

**Reviewer Scores:**

- y5pj: 6 to 8 (as it's mentioned, they raised after rebuttal)

- 6oCC: 4 to 6

- mKuR: 2 -> remain at 2 or increase to 4

---

### Decision · Program_Chairs · 2026-01-26

Accept (Poster)